# Guiding Instruction-based Image Editing via Multimodal Large Language Models

 Tsu-Jui Fu[1], Wenze Hu[2], Xianzhi Du[2], William Yang Wang[1], Yinfei Yang[2], Zhe Gan[2]
[1]UC Santa Barbara, [2]Apple

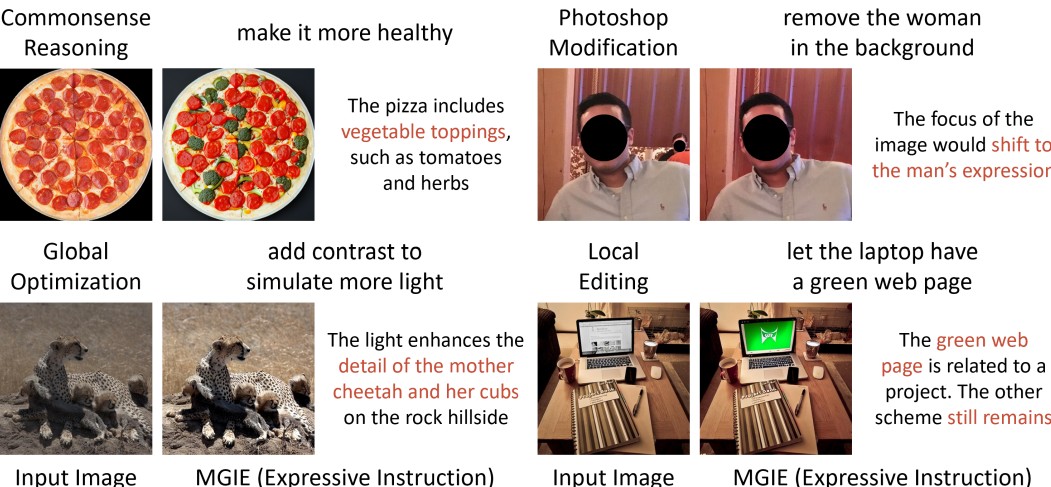

Figure 1: We introduce MLLM-Guided Image Editing (MGIE) to improve instruction-based image editing for various editing aspects. The top is the input instruction, and the right is the jointly derived expressive instruction by MGIE.

## Abstract

Instruction-based image editing improves the controllability and flexibility of image manipulation via natural commands without elaborate descriptions or regional masks. However, human instructions are sometimes too brief for current methods to capture and follow. Multimodal large language models (MLLMs) show promising capabilities in cross-modal understanding and visual-aware response generation via LMs. We investigate how MLLMs facilitate edit instructions and present MLLM-Guided Image Editing (MGIE). MGIE learns to derive expressive instructions and provides explicit guidance. The editing model jointly captures this visual imagination and performs manipulation through end-to-end training. We evaluate various aspects of Photoshop-style modification, global photo optimization, and local editing. Extensive experimental results demonstrate that expressive instructions are crucial to instruction-based image editing, and our MGIE can lead to a notable improvement in automatic metrics and human evaluation while maintaining competitive inference efficiency.

## 1 Introduction

Visual design tools are widely adopted in various multimedia fields nowadays. Despite considerable demand, they require prior knowledge to operate. To enhance controllability and accessibility, text-guided image editing has obtained popularity in recent studies (Li et al., 2020; Patashnik et al., 2021; Crowson et al., 2022; Gal et al., 2022). With an attractive ability to model realistic images, diffusion models (Ho et al., 2020) are also adopted in image editing (Kim et al., 2022). By swapping the latent cross-modal maps, models can perform visual manipulation to reflect the alteration of the input-goal

---

 Work done during an internship at Apple. Project website: https://mllm-ie.github.io

caption (Hertz et al., 2023; Mokady et al., 2022; Kawar et al., 2023). They can further edit a specific region by a guided mask (Nichol et al., 2022; Avrahami et al., 2022). Instead of relying on elaborate descriptions or regional masks, instruction-based editing (El-Nouby et al., 2019; Li et al., 2020; Fu et al., 2020) allows human commands that directly express how and which aspect of an image to edit. This flexibility also benefits practicality as such guidance is more aligned with human intuition.

Due to the data scarcity of the input-goal-instruction triplet, InsPix2Pix (Brooks et al., 2023) collects a curated IPr2Pr dataset. The instruction is generated by GPT-3 (Brown et al., 2020), and the input-goal image pair is synthesized from Prompt-to-Prompt (Hertz et al., 2023). InsPix2Pix then applies a pre-trained CLIP text encoder (Radford et al., 2021) to lead the diffusion model along with the input image. Although having feasible results, CLIP is trained for static descriptions, which is challenging to capture the essential visual transformation in editing. Furthermore, the instruction is too brief but ambiguous and insufficient to guide toward the intended goal. The deficiency limits the effectiveness of InsPix2Pix in instruction-based image editing.

Large language models (LLMs) (Brown et al., 2020; Touvron et al., 2023) have shown significant advancement in diverse language tasks, including machine translation, text summarization, and question answering. Learning from large-scale corpora with diverse content, LLMs contain latent visual knowledge and creativity, which can assist various vision-and-language tasks (Wu et al., 2023; Feng et al., 2023; Chakrabarty et al., 2023). Upon LLMs, multimodal large language models (MLLMs) can treat images as input naturally and provide visual-aware responses to serve as multimodal assistants (Zhang et al., 2023b; Liu et al., 2023; Zhu et al., 2023; Koh et al., 2023).

Inspired by MLLMs, we incorporate them to deal with the insufficient guidance issue of instructions and introduce MLLM-Guided Image Editing (MGIE). As demonstrated in Fig. 2, MGIE consists of an MLLM and a diffusion model. The MLLM learns to derive concise expressive instructions and offers explicit visual-related guidance. The diffusion model is jointly updated and performs image editing with the latent imagination of the intended goal via end-to-end training. In this way, MGIE benefits from the inherent visual derivation and addresses ambiguous human commands to achieve reasonable editing. For the example in Fig. 1, it is difficult to capture what "*healthy*" means without additional context. Our MGIE can precisely connect "*vegetable toppings*" with the pizza and lead to the related editing as human expectation.

To learn instruction-based image editing, we adopt IPr2Pr as our pre-training dataset. We consider different editing aspects in EVR (Tan et al., 2019), GIER (Shi et al., 2020), MA5k (Shi et al., 2022), and MagicBrush (Zhang et al., 2023a). MGIE performs Photoshop-style modification, global photo optimization, and local object alteration. All should be guided by human instructions. Experimental results indicate that our MGIE significantly strengthens instruction-based image editing with reasonable expressive instructions in automatic metrics and human evaluation, and visual-aware guidance is crucial to this improvement. In summary, our contributions are three-fold:

- We introduce MLLM-Guided Image Editing (MGIE), which jointly learns the MLLM and editing model with visual-aware expressive instructions to provide explicit guidance.
- We conduct comprehensive studies from various editing aspects, including Photoshop-style modification, global photo optimization, and local editing, along with qualitative comparisons.
- Extensive experiments demonstrate that visual-aware expressive instructions are crucial for image editing, and our MGIE effectively enhances editing performance.

## 2 RELATED WORK

**Instruction-based Image Editing.** Text-guided image editing can significantly improve the controllability and accessibility of visual manipulation by following human commands. Previous works built upon the GAN frameworks (Goodfellow et al., 2015; Reed et al., 2016) to alter images but are limited to unrealistic synthesis or specific domains (Nam et al., 2018; Li et al., 2020; El-Nouby et al., 2019; Fu et al., 2020; 2022). With promising large-scale training, diffusion models (Ho et al., 2020; Ramesh et al., 2022; Sahari et al., 2022; Rombach et al., 2022) can accomplish image transformation via controlling the cross-modal attention maps for the global caption (Meng et al., 2022; Hertz et al., 2023; Kawar et al., 2023; Gu et al., 2023). Local image editing allows fine-grained manipulation by inpainting target regions with user-provided (Nichol et al., 2022; Avrahami et al., 2022; Wang et al., 2023b) or predicted masks (Bar-Tal et al., 2022; Couairon et al., 2023) while preserving the remain-

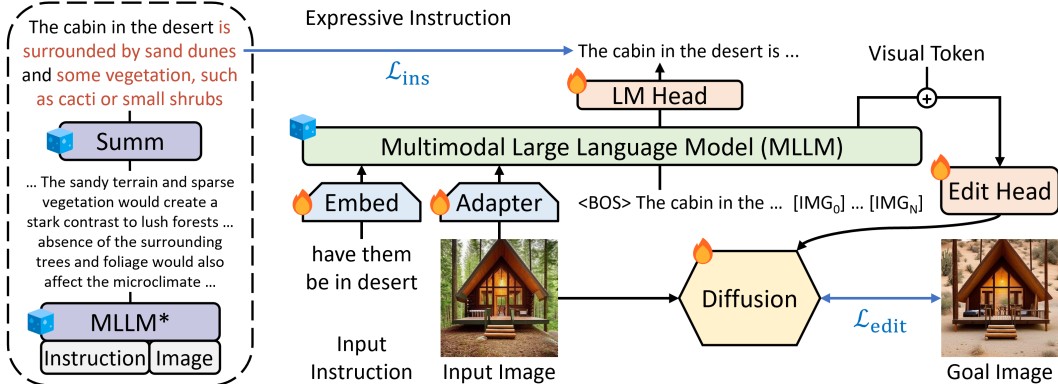

Figure 2: Overview of MLLM-Guided Image Editing (**MGIE**), which leverages MLLMs to enhance instruction-based image editing. MGIE learns to derive concise expressive instructions and provides explicit visual-related guidance for the intended goal. The diffusion model jointly trains and achieves image editing with the latent imagination through the edit head in an end-to-end manner. 🔥 and 💠 show the module is trainable and frozen[1], respectively.

ing areas. Different from them, instruction-based image editing accepts straight commands, such as "*add fireworks to the sky*", which is not restricted to elaborate descriptions or regional masks. Recent methods learn from synthetic input-goal-instruction triples (Brooks et al., 2023) and with additional human feedback (Zhang et al., 2023c) to follow editing instructions. However, the frozen CLIP text encoder is pre-trained for static descriptions but not the crucial transformation in editing. Moreover, the instructions are sometimes ambiguous and imprecise for the editing goal. In this paper, we learn with multimodal large language models to perceive images along with given prompts for expressive instructions, which provides explicit yet detailed guidance, leading to superior editing performance.

**Large Language Models for Vision.** Large language models (LLMs) have demonstrated impressive capabilities for text generation and generalizability in various tasks (Brown et al., 2020; Chowdhery et al., 2022; Touvron et al., 2023). With robust text understanding, previous works adapt LLMs for input prompts and reason downstream vision-and-language tasks (Zhang et al., 2023d; Wu et al., 2023; Lu et al., 2023; Yang et al., 2023; Chakrabarty et al., 2023). They further produce pseudocode instructions or executable programs by LLMs (Huang et al., 2022; Gupta & Kembhavi, 2023; Surís et al., 2023; Feng et al., 2023; Lian et al., 2023). Through visual feature alignment with instruction tuning, multimodal large language models (MLLMs) can perceive images and provide adequate responses (Li et al., 2023b; Zhang et al., 2023b; Liu et al., 2023; Zhu et al., 2023). Recently, studies also adopt MLLMs for generating chat-related images (Koh et al., 2023; Sun et al., 2023). However, they can only produce images from scratch, which are distinct from inputs. Our proposed MGIE is the first to leverage MLLMs and improve image editing with derived expressive instructions.

## 3 METHOD

### 3.1 BACKGROUND: MULTIMODAL LARGE LANGUAGE MODELS (MLLMS)

Large language models (LLMs) have shown impressive capabilities for natural language generation. Multimodal large language models (MLLMs) empower LLMs to perceive images and provide reasonable responses. Initialized from a pre-trained LLM, the MLLM contains a visual encoder (*e.g.,* CLIP-L (Radford et al., 2021)) to extract the visual features $f$, and an adapter $\mathcal{W}$ to project $f$ into the language modality. We follow the training of LLaVA (Liu et al., 2023), which is summarized as:

$$
\begin{aligned}
\mathcal{C} &= \{x_1, x_2, ..., x_l\}, \\
f &= \text{Enc}_{\text{vis}}(\mathcal{V}), \\
x_t &= \text{MLLM}(\{x_1, ...x_{t-1}\} \mid \mathcal{W}(f)),
\end{aligned}
\tag{1}
$$

---

[1]We adopt Flan-T5-XXL (Chung et al., 2022), which has been specifically fine-tuned for summarization, as our summarization model for the original MLLM (MLLM*).

where $l$ is the length of the word token in $\mathcal{C}$. $\mathcal{C}$ can be the image caption (Features Alignment) or the multimodal instruction-following data (Instruction Tuning). The MLLM follows the standard auto-regressive training for the next token prediction and then can serve as a visual assistant for various tasks such as visual question answering and complex reasoning. Although the MLLM is capable of visual perception via the above training, its output is still limited to text.

## 3.2 MLLM-GUIDED IMAGE EDITING (MGIE)

As illustrated in Fig. 2, we propose MLLM-Guided Image Editing (MGIE) to edit an input image $\mathcal{V}$ into a goal image $\mathcal{O}$, by a given instruction $\mathcal{X}$. To handle imprecise instructions, MGIE contains the MLLM and learns to derive explicit yet concise expressive instructions $\mathcal{E}$. To bridge the language and visual modality, we add special [IMG] tokens after $\mathcal{E}$ and adopt the edit head $\mathcal{T}$ to transform them. They serve as the latent visual imagination from the MLLM and guide our diffusion model $\mathcal{F}$ to achieve the intended editing goal. MGIE is then able to comprehend ambiguous commands with visual-related perception for reasonable image editing.

**Concise Expressive Instruction.** From features alignment and instruction tuning, the MLLM can offer visual-related responses with its cross-modal perception. For image editing, we use this prompt "*what will this image be like if* [instruction]" as the language input with the image and derive a detailed explanation of the editing command. However, those explanations are always too lengthy and involve redundant descriptions, which even mislead the intention. To obtain succinct narrations, we apply a pre-trained summarizer [1] and make the MLLM learn to generate the summarized outputs. We treat this explicit yet concise guidance as expressive instruction $\mathcal{E}$:

$$
\begin{aligned}
\mathcal{E} &= \text{Summ}(\text{MLLM*}([\text{prompt}, \mathcal{X}] \mid \mathcal{W}(f))) \\
&= \{w_1, w_2, ..., w_l\}, \\
w'_t &= \text{MLLM}(\{w_1, ..., w_{t-1}\} \mid \mathcal{W}(f)), \\
\mathcal{L}_{\text{ins}} &= \sum\nolimits_{t=1}^{l} \text{CELoss}(w'_t, w_t),
\end{aligned}
\tag{2}
$$

where we apply the cross-entropy loss (CELoss) to train the MLLM via teacher forcing. $\mathcal{E}$ can provide a more concrete idea than $\mathcal{X}$ such as linking "*dessert*" with "*sand dunes*" and "*cacti or small shrubs*", which mitigates the comprehension gap for reasonable image editing. This strategy further enhances our efficiency. During inference, the trained MGIE straightforwardly derives concise $\mathcal{E}$ instead of rolling out lengthy narrations (22.7 *vs.* 64.5 tokens) and relying on external summarization. MGIE now can acquire a visual imagination of the editing intention but is confined to the language modality. To bridge the gap, we append $N$ visual tokens [IMG] after $\mathcal{E}$, where their word embeddings are trainable, and the MLLM also learns to generate them through its language modeling (LM) head. Inspired by GILL (Koh et al., 2023), the visual tokens are treated as visual-related instruction understanding in $\mathcal{E}$ and establish a connection between the language and vision modalities.

**Image Editing via Latent Imagination.** We adopt the edit head $\mathcal{T}$ to transform [IMG] into actual visual guidance. $\mathcal{T}$ is a sequence-to-sequence model, which maps the sequential visual tokens from the MLLM to the semantically meaningful latent $\mathcal{U} = \{u_1, u_2, ..., u_L\}$ as the editing guidance:

$$
u_t = \mathcal{T}(\{u_1, ..., u_{t-1}\} \mid \{e_{[\text{IMG}]} + h_{[\text{IMG}]}\}),
\tag{3}
$$

where $e$ is the word embedding and $h$ is the hidden state (from the last layer of MLLM before the LM head) of [IMG]. Specifically, the transformation over $e$ can be treated as a general representation in the visual modality, and $h$ is an instance-aware visual imagination for such editing intention. Our $\mathcal{T}$ is similar to GILL and BLIP-2 (Li et al., 2023b;a) for extracting visual features.

To guide image editing with the visual imagination $\mathcal{U}$, we consider a latent diffusion model $\mathcal{F}$ (Rombach et al., 2022), which includes the variational autoencoder (VAE) and addresses denoising diffusion in the latent space. Our goal of $\mathcal{F}$ is to generate the latent goal $o = \text{Enc}_{\text{VAE}}(\mathcal{O})$ from preserving the latent input $v = \text{Enc}_{\text{VAE}}(\mathcal{V})$ and following the editing guidance $\{u\}$. The diffusion process keeps adding noises to $o$ as $z_t$, where the noise level is increasing over timesteps $t$. We then learn the UNet $\epsilon_\theta$ to predict the added noise (Ho et al., 2020). As LDM, we inject the visual imagination into $\epsilon_\theta$ via the cross-attention layer $\text{Attention}(Q, K, V) = \text{softmax}(\frac{QK^T}{\sqrt{\dim}}) \cdot V$ with

$$
Q = W_Q^{(i)} \cdot \varphi_i(z_t), K = W_K^{(i)} \cdot \{u\}, V = W_V^{(i)} \cdot \{u\},
\tag{4}
$$

| Method | EVR | | | GIER | | | MA5k | | | MagicBrush | | | |
|---|---|---|---|---|---|---|---|---|---|---|---|---|---|
| | L1↓ | DINO↑ | CVS↑ | L1↓ | SSIM↑ | CVS↑ | L1↓ | SSIM↑ | LPIPS↓ | L1↓ | DINO↑ | CVS↑ | CTS↑ |
| InsPix2Pix | 0.189 | 67.82 | 81.38 | 0.144 | 57.51 | 86.63 | 0.176 | 58.92 | 0.359 | 0.101 | 71.46 | 85.22 | 29.34 |
| LGIE | **0.159** | 69.71 | **82.04** | 0.152 | 56.86 | 86.99 | 0.144 | 64.60 | 0.327 | 0.084 | 80.90 | 88.87 | 30.10 |
| MGIE | 0.163 | **71.49** | 81.73 | **0.135** | **59.24** | **88.59** | **0.133** | **66.25** | **0.298** | **0.082** | **82.22** | **91.14** | **30.40** |

Table 1: **Zero-shot editing results**. All models are only pre-trained on IPr2Pr (Brooks et al., 2023).

| Method | EVR | | | GIER | | | MA5k | | | MagicBrush | | | |
|---|---|---|---|---|---|---|---|---|---|---|---|---|---|
| | L1↓ | DINO↑ | CVS↑ | L1↓ | SSIM↑ | CVS↑ | L1↓ | SSIM↑ | LPIPS↓ | L1↓ | DINO↑ | CVS↑ | CTS↑ |
| InsPix2Pix | 0.166 | 70.79 | 82.76 | 0.111 | 64.86 | 91.49 | 0.122 | 67.12 | 0.267 | 0.063 | 87.99 | 93.83 | 30.93 |
| LGIE | 0.147 | 74.71 | 85.06 | **0.104** | 65.30 | 90.61 | 0.094 | 71.47 | 0.246 | 0.058 | 88.09 | 93.57 | 31.33 |
| MGIE | **0.146** | **75.65** | **85.28** | 0.105 | **68.68** | **92.42** | **0.082** | **72.91** | **0.235** | **0.057** | **90.65** | **95.28** | **31.73** |

Table 2: **Fine-tuned editing results**. All models are further fine-tuned and adapted to each dataset.

where $\varphi$ is the flattened operation, $W_Q^{(i)}$, $W_K^{(i)}$, and $W_V^{(i)}$ are learnable attention matrices. Following InsPix2Pix, we also concatenate $v$ with $z_t$. In this way, our $\mathcal{F}$ can condition both $\mathcal{V}$ and $\mathcal{U}$ to perform image editing. We take classifier-free guidance (Ho & Salimans, 2021), and the score estimation $s_\theta$ is extrapolated to keep away from the unconditional $\varnothing$, where the editing loss $\mathcal{L}_{\text{edit}}$ is calculated as:

$$
\begin{aligned}
s_\theta(z_t, v, \{u\}) = \; & s_\theta(z_t, \varnothing, \varnothing) \\
& + \alpha_{\mathcal{V}} \cdot (s_\theta(z_t, v, \varnothing) - s_\theta(z_t, \varnothing, \varnothing)) \\
& + \alpha_{\mathcal{X}} \cdot (s_\theta(z_t, v, \{u\}) - s_\theta(z_t, v, \varnothing)), \\
\mathcal{L}_{\text{edit}} = \; & \mathbb{E}_{o,v,\{u\},\epsilon \sim \mathcal{N}(0,1),t} \left[ ||\epsilon - \epsilon_\theta(z_t, t, v, \{u\})||_2^2 \right],
\end{aligned}
\tag{5}
$$

where $\alpha_{\mathcal{V}}$ and $\alpha_{\mathcal{X}}$ are the weights of the guidance scale for the image and the instruction. Similar to InsPix2Pix, we randomly make $v = \varnothing$, $\{u\} = \varnothing$, or both $= \varnothing$ for 5% of data during training. After we have the generated latent $o'$ through the denoising process by $\epsilon_\theta$, we can obtain the editing result $O' = \text{Dec}_{\text{VAE}}(o')$. During inference, we use $\alpha_{\mathcal{V}} = 1.5$ and $\alpha_{\mathcal{X}} = 7.5$.

### 3.3 LEARNING OF MGIE

Algo. 1 presents the learning process of the proposed MGIE. The MLLM learns to derive concise $\mathcal{E}$ via the instruction loss $\mathcal{L}_{\text{ins}}$. With the latent imagination from [IMG], $\mathcal{T}$ transforms their modality and guides $\mathcal{F}$ to synthesize the resulting image. The editing loss $\mathcal{L}_{\text{edit}}$ is applied for diffusion training. Most weights can be frozen (self-attention blocks inside the MLLM), leading to parameter-efficient end-to-end training. Overall optimization of $\mathcal{L}_{\text{all}} = \mathcal{L}_{\text{ins}} + 0.5 \cdot \mathcal{L}_{\text{edit}}$ can be:

$$
\min_{\text{MLLM},\mathcal{W},\mathcal{T},\mathcal{F}} \mathcal{L}_{\text{all}} . \tag{6}
$$

---

**Algorithm 1** MLLM-Guided Image Editing

1: **while** TRAIN_MGIE **do**
2:     $\mathcal{V}, \mathcal{X}, \mathcal{O} \leftarrow$ input/instruction/goal triple
3:     $\{w\} \leftarrow$ summarized explanation
4:     $\{w'\} = \text{MLLM}(\mathcal{V} \mid \mathcal{X})$
5:     $\mathcal{L}_{\text{ins}} \leftarrow$ instruction loss       ▷ Eq. 2
6:     $\mathcal{U} = \mathcal{T}(\{[\text{IMG}]\})$
7:     $\mathcal{O}' = \mathcal{F}(\mathcal{V}, \mathcal{U})$
8:     $\mathcal{L}_{\text{edit}} \leftarrow$ editing loss       ▷ Eq. 5
9:     $\mathcal{L}_{\text{all}} \leftarrow$ overall training loss
10: **end while**

---

## 4 EXPERIMENTS

### 4.1 EXPERIMENTAL SETUP

**Datasets and Evaluation Metrics.** We use **IPr2Pr** (Brooks et al., 2023) as our pre-training data. It contains 1M CLIP-filtered data, where instructions are extracted by GPT-3 (Brown et al., 2020), and images are synthesized by Prompt-to-Prompt (Hertz et al., 2023). For a comprehensive evaluation, we consider various editing aspects. **EVR** (Tan et al., 2019) collects 5.7K triples from PhotoshopRequest. We treat the standard pixel difference (L1) and visual feature similarity from DINO (Caron et al., 2021) or the CLIP visual encoder (CVS) between generated images and ground-truth goals as the evaluation metrics. **GIER** (Shi et al., 2020) crawls a larger-scale 29.9K triples also from online forums. Since there are more examples about global optimization, we apply L1, CVS, and Structural

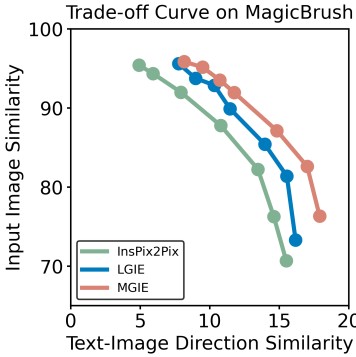

Figure 3: **Trade-off curve for image editing**. We set $\alpha_{\mathcal{X}}$ as 7.5 and vary $\alpha_{\mathcal{V}}$ in $[1.0, 2.2]$. For both edit (X-axis) and input consistency (Y-axis), higher is better.

| Arch. | Method | MA5k | | | MagicBrush | | | |
|---|---|---|---|---|---|---|---|---|
| | | L1↓ | SSIM↑ | LPIPS↓ | L1↓ | DINO↑ | CVS↑ | CTS↑ |
| FZ | InsPix2Pix | 0.176 | **58.92** | **0.359** | **0.101** | 71.46 | 85.22 | 29.34 |
| | LGIE | 0.178 | 57.26 | 0.372 | 0.133 | 67.53 | 82.49 | 28.79 |
| | MGIE | **0.163** | 57.54 | 0.366 | 0.128 | **71.65** | **86.00** | **29.43** |
| FT | LGIE | 0.166 | 60.11 | 0.357 | 0.124 | 71.04 | 85.47 | 29.37 |
| | MGIE | **0.163** | **61.38** | **0.348** | **0.101** | **74.79** | **87.12** | **29.68** |
| E2E | LGIE | 0.144 | 64.60 | 0.327 | 0.084 | 80.90 | 88.87 | 30.10 |
| | MGIE | **0.133** | **66.25** | **0.298** | **0.082** | **82.22** | **91.14** | **30.40** |

Table 3: **Ablation study**. We attempt FZ, FT, or E2E to utilize expressive instructions. **FZ** directly treats expressive instructions as the inputs to frozen InsPix2Pix. **FT** further fine-tunes InsPix2Pix and makes it adapt to expressive instructions. Our **E2E** learns expressive instructions along with the MLLM and trains the diffusion model in an end-to-end manner.

Similarity Index (SSIM). **MA5k** (Shi et al., 2022) consists of 24.8K triples and aims at changing the contrast, brightness, or saturation of a whole photo. We leverage L1, SSIM, and Learned Perceptual Image Patch Similarity (LPIPS) (Zhang et al., 2018) as the photo difference[2]. **MagicBrush** (Zhang et al., 2023a) annotates 10.5K triples. We follow them to use L1, DINO, CVS, and text-visual feature similarity (CTS) (Hessel et al., 2021) between goal captions and resulting images. We treat the same training/validation/testing split as the original settings. Without specific mention, all evaluations are averaged from 5 random seeds in a zero-shot manner, where models are only trained on IPr2Pr.

**Baselines.** We treat InsPix2Pix (Brooks et al., 2023), built upon the CLIP text encoder with a diffusion model for instruction-based image editing, as our baseline. We consider a similar LLM-guided image editing (LGIE) model, where LLaMA-7B (Touvron et al., 2023) is adopted for expressive instructions $\mathcal{E}$ from instruction-only inputs but without visual perception.

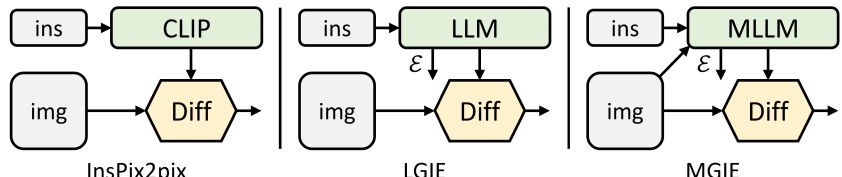

**Implementation Details.** The MLLM and diffusion model $\mathcal{F}$ are initialized from LLaVA-7B (Liu et al., 2023) and StableDiffusion-v1.5 (Rombach et al., 2022). We jointly update both for the image editing task. Note that only word embeddings and LM head in the MLLM are trainable. Following GILL (Koh et al., 2023), we use $N$=8 visual tokens. The edit head $\mathcal{T}$ is a 4-layer Transformer, which transforms language features into editing guidance. We adopt AdamW (Loshchilov & Hutter, 2019) with the batch size of 128 to optimize MGIE. The learning rates of the MLLM and $\mathcal{F}$ are 5e-4 and 1e-4, respectively. All experiments are conducted in PyTorch (Paszke et al., 2017) on 8 A100 GPUs.

## 4.2 QUANTITATIVE RESULTS

Table 1 shows the zero-shot editing results, where models are trained only on IPr2Pr. For EVR and GIER that involve Photoshop-style modifications, expressive instructions can reveal concrete goals instead of brief but ambiguous commands, which makes the editing results more similar to intentions (*e.g.*, higher 82.0 CVS on EVR by LGIE and higher 59.2 SSIM on GIER by MGIE). For global photo optimization on MA5k, InsPix2Pix is hard to deal with due to the scarcity of related training triples. Though trained from the same source, LGIE and MGIE can offer detailed explanations via learning with the LLM, but LGIE is still confined to its single modality. With access to images, MGIE derives explicit instructions such as *which regions should brighten* or *what objects are more distinct*. It can bring a significant performance boost (*e.g.*, higher 66.3 SSIM and lower 0.3 photo distance). Similar

---

[2]As there is no object alteration in MA5k, feature-based DINO and CVS cannot clearly tell the difference.

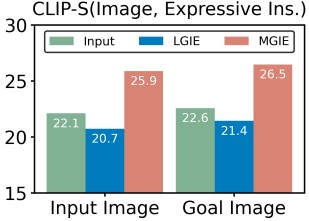

Figure 4: **CLIP-S** across images (input / goal) and expressive instructions.

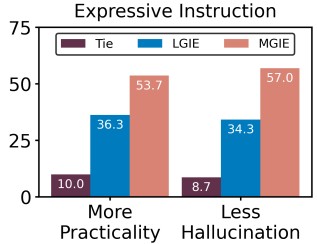

Figure 5: **Human eval** of expressive instructions quality.

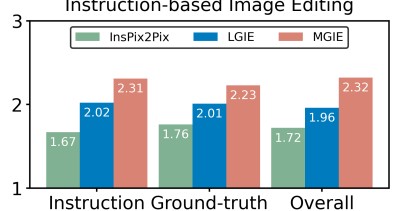

Figure 6: **Human eval** of image editing results in terms of instruction following, ground-truth relevance, and overall quality.

results are found on MagicBrush. MGIE also achieves the best performance from the precise visual imagination and modifies the designate targets as the goals (*e.g.*, higher 82.2 DINO visual similarity and higher 30.4 CTS global caption alignment).

To investigate instruction-based image editing for the specific purpose, Table 2 fine-tunes models on each dataset. For EVR and GIER, all models obtain improvements after the adaptation to Photoshop-style editing tasks. Since fine-tuning makes expressive instructions more domain-specific as well, our MGIE increases the most via learning with domain-related guidance. This also helps our diffusion model to demonstrate concrete edited scenes from the fine-tuned MLLM, which benefits both global optimization and local modification (*e.g.*, notably lower 0.24 LPIPS on MA5k and higher 95.3 CVS on MagicBrush). MGIE is consistently superior to LGIE in all aspects of editing since our visual-aware guidance is more aligned with the intended goal. From the above experiments, we illustrate that learning with expressive instructions can effectively enhance image editing, and visual perception plays a crucial role in deriving explicit guidance for the greatest enhancements.

**Trade-off between $\alpha_\mathcal{X}$ and $\alpha_\mathcal{V}$.** There are two goals in image editing: manipulate the target as the instruction and preserve the remaining as the input image. Fig. 3 plots the trade-off curves between the instruction ($\alpha_\mathcal{X}$) and input consistency ($\alpha_\mathcal{V}$). We fix $\alpha_\mathcal{X}$ as 7.5 and vary $\alpha_\mathcal{V}$ in $[1.0, 2.2]$. Higher $\alpha_\mathcal{V}$ will make an editing result more similar to the input but less aligned with the instruction. X-axis calculates the CLIP directional similarity as how much the editing follows the instruction; Y-axis is the feature similarity to the input image from the CLIP visual encoder. Through concrete expressive instructions, we surpass InsPix2Pix in all settings. Our MGIE additionally results in comprehensive enhancements by learning with explicit visual-related guidance. This supports robust improvement, whether requiring higher input correlation or edit relevance.

### 4.3 Ablation Study

MLLM-Guided Image Editing exhibits encouraging improvement in both zero-shot and fine-tuning scenarios. Now, we investigate different architectures to use expressive instructions. Table 3 considers **FZ**, **FT**, and our **E2E**. FZ directly uses the derived expressive instructions[3] as the input prompts to the frozen InsPix2Pix. In spite of having additional guidance, the scenario still differs from the trained editing instructions, which makes it difficult to deal with. LGIE even hurts the performance as it may mislead due to the shortage of visual perception. FT fine-tunes InsPixPix and adapts it to expressive instructions. These results support that image editing can benefit from explicit guidance along the derivation of instructions from the LLM/MLLM. E2E updates the editing diffusion model in conjunction with the LM, which learns to extract applicable guidance and discard irrelevant narration simultaneously through the end-to-end hidden states. In addition, our E2E can also avoid the potential error that may be propagated from the expressive instructions. Hence, we observe the most enhancements in both global optimization (MA5k) and local editing (MagicBrush). Among FZ, FT, and E2E, MGIE consistently surpasses LGIE. This indicates that expressive instructions with crucial visual perception are always advantageous across all ablation settings.

**Why MLLM Guidance is Helpful?** Fig. 4 presents the CLIP-Score between input or ground-truth goal images and expressive instructions. A higher CLIP-S to input images indicates that instructions

---

[3]During the ablation study, we employ concise summarized expressive instructions for a fair comparison.

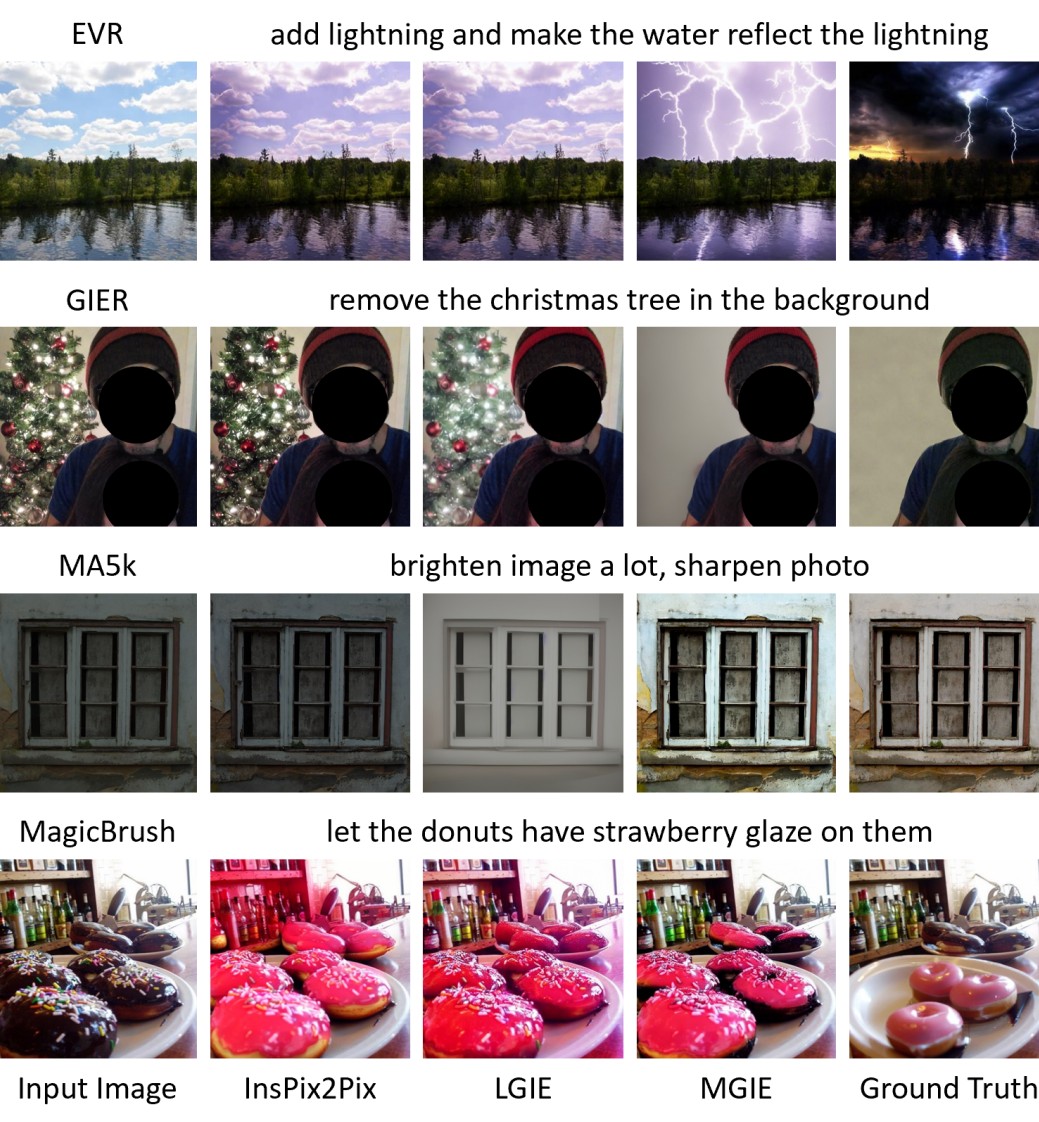

Figure 7: **Qualitative comparison** between InsPix2Pix, LGIE, and our MGIE. For the 1st example, MGIE can showcase the clear "*lightning*" in the sky and its reflection on the water. For the 2nd one, although LGIE accurately targets the Christmas tree, only MGIE removes it in the background. For photo optimization (the 3rd example), InsPix2Pix fails to adjust the brightness, and LGIE makes the whole photo white and obviously distinct. In contrast, MGIE follows the instruction to brighten as well as sharpen it. Moreover, in the 4th one, MGIE puts the "*glaze*" only on the donuts, but baselines even draw the entire image in strawberry pink.

are relevant to the editing source. Better alignment with goal images provides explicit and correlated edit guidance. Without access to visual perception, expressive instructions from LGIE are limited to general language imagination, which is not tailored to the source image. The CLIP-S are even lower than the original instructions. By contrast, MGIE is more aligned with inputs/goals, which explains why our expressive instructions are helpful. With a clear narration of the intended result, our MGIE can achieve the greatest improvements in image editing.

**Human Evaluation.** Apart from automatic metrics, we conduct a human evaluation to study generated expressive instructions and image editing results. We randomly sample 25 examples for each dataset (100 in total) and consider humans to rank across baselines and MGIE. To avoid potential ranking bias, we hire 3 annotators for each example. Fig. 5 plots the quality of generated expressive instructions. Precise guidance is informative and aligns with the intended goal (More Practicality). At the same time, it should avoid incorrect or unrelated explanations (Less Hallucination). Firstly,

| EVR | change the background and add a star wars background | GIER | remove all people except baby from image |
|---|---|---|---|
| 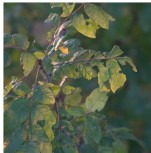 | If you were to change the background to Star Wars movie, the image would look like a Star Trek background. / Add a background that is related to the Star War franchise, such as a lightsaber or a spaceship. | 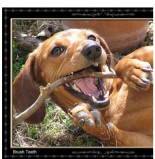 | The only people still in the image would be the two people in the foreground. The rest needs to be removed. / If we remove all, we would be left with a baby sitting on the beach with a shark and a few balloons. |

EVR — change the background and add a star wars background

If you were to change the background to Star Wars movie, the image would look like a Star Trek background.

Add a background that is related to the Star War franchise, such as a lightsaber or a spaceship.

GIER — remove all people except baby from image

The only people still in the image would be the two people in the foreground. The rest needs to be removed.

If we remove all, we would be left with a baby sitting on the beach with a shark and a few balloons.

MA5k — the desired image has more light contrast

Adjust the contrast and brightness of a desired area, making it look darker and more visible.

Adjust between the light and dark areas to bring out the details of the leaves and the tree trunk.

MagicBrush — close the dog's mouth

Take a picture of a dog-bitten object with the dog in the foreground and a dark background.

If the dog in the image had its mouth covered, it would appear as if it were "playing with" the stick or chewing on it.

Input Image   LGIE   MGIE          Input Image   LGIE   MGIE

Figure 8: **Qualitative comparison** of expressive instructions by LGIE and our MGIE. Due to the limitation of the single modality, LGIE can only have language-based insight but may derive irrelevant or even wrong explanations for image editing (*e.g.*, "*two people still in the foreground*" for GIER). With access to images, MGIE provides explicit visual imagination after the editing such as "*baby on the beach with a shark*" or "*bring out details of leaves and trunk*". More surprisingly, we can link "*lightsaber or spaceship*" from Star Wars and describe "*chewing on the stick*" for the dog, which is aligned with the intended goal.

over 53% support that MGIE provides more practical expressive instructions, which facilitates the image editing task with explicit guidance. Meanwhile, 57% of labelers indicate that our MGIE can prevent irrelevant descriptions from language-derived hallucinations in LGIE since it perceives the image to have a precise goal for editing. Fig. 6 compares the image editing results by InsPix2Pix, LGIE, and our MGIE in terms of instruction following, ground-truth relevance, and overall quality. The ranking score is ranging from 1 to 3, higher is better. With derived expressive instructions from the LLM or MLLM, LGIE and MGIE both outperform the baseline and perform image editing that is correlated with the instruction as well as similar to the ground-truth goal. Additionally, since our expressive instructions can provide concrete and visual-aware guidance, MGIE has the best human preference in all aspects, including the overall editing quality. These performance trends also align with automatic evaluations, which support our usage of metrics.

**Inference Efficiency.** Despite relying on MLLM to facilitate image editing, MGIE only rolls out concise expressive instructions (less than 32 tokens) and contains feasible efficiency as InsPix2Pix. Table 4 presents the inference time cost on an NVIDIA A100 GPU. For a single input, MGIE can accomplish the editing task in 10 seconds. With greater data parallelization, we take a similar amount of time (*e.g.*, 37 seconds when batch size 8). The entire process can be affordable in one GPU (40GB). In summary, our MGIE surpasses the baseline on quality yet maintains competitive efficiency, leading to effective and practical image editing.

| BS | InsPix2Pix | MGIE |
|---|---|---|
| 1 | 6.8 | 9.2 |
| 4 | 16.5 | 20.6 |
| 8 | 31.5 | 36.9 |

Table 4: Time cost.

**Qualitative Comparisons.** Fig. 7 illustrates the visualized comparison on all used datasets. Fig. 8 further compares the expressive instructions by LGIE or MGIE. Our superior performance benefits from the explicit guidance of visual-related expressive instructions. Please visit our project website[4] for more qualitative results.

## 5 CONCLUSION

We propose MLLM-Guided Image Editing (MGIE) to enhance instruction-based image editing via learning to produce expressive instructions. Instead of brief but ambiguous guidance, MGIE derives explicit visual-aware intention and leads to reasonable image editing. We conduct extensive studies from various editing aspects and demonstrate that our MGIE effectively improves performance while maintaining competitive efficiency. We also believe the MLLM-guided framework can contribute to future vision-and-language research.

---

[4]Project website: `https://mllm-ie.github.io`

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
