| Method | MA5k | | MagicBrush | | |
|---|---|---|---|---|---|
| | SSIM↑ | LPIPS↓ | DINO↑ | CVS↑ | CTS↑ |
| InsPix2Pix | 58.92 | 0.359 | 71.46 | 85.22 | 29.34 |
| + Enc$_{LLaMA}$ | 59.08 | **0.334** | 72.38 | 85.99 | 29.29 |
| + Enc$_{LLaVA}$ | **60.94** | 0.352 | **74.10** | **87.21** | **29.37** |
| HIVE | 65.17 | 0.302 | 78.95 | 88.23 | 29.42 |
| InsEdit | 59.59 | 0.364 | **83.26** | **91.16** | 29.80 |
| MGIE | **66.25** | **0.298** | 82.22 | 91.14 | **30.40** |

Table 5: **Zero-shot editing comparison** to different instruction encoders (Enc), human feedback (HIVE), and mask-then-inpaint (InsEdit).

| Method | Size | MA5k | | MagicBrush | | |
|---|---|---|---|---|---|---|
| | | SSIM↑ | LPIPS↓ | DINO↑ | CVS↑ | CTS↑ |
| InsPix2Pix | | 58.92 | 0.359 | 71.46 | 85.22 | 29.34 |
| LGIE | 7B | **64.60** | 0.327 | **80.90** | **88.87** | 30.10 |
| | 13B | 63.50 | **0.308** | 80.18 | 88.77 | **30.31** |
| MGIE | 6.7B | 63.78 | 0.300 | 78.82 | 90.01 | 29.47 |
| | 7B | **66.25** | 0.298 | **82.22** | 91.14 | 30.40 |
| | 13B | 65.91 | **0.279** | 82.15 | **91.52** | **30.75** |

Table 6: **Zero-shot editing comparison** of different LM sizes. We treat the visual-tuned OPT-6.7B in our used MGIE-6.7B.

## A ADDITIONAL RESULTS

**Comparison to More Baselines.** InsPix2Pix (Brooks et al., 2023) applies the CLIP encoder (Radford et al., 2021), which is insufficient to capture the transformation for editing. We treat the stronger LLM/MLLM as the instruction encoder (Enc) and follow the same training strategy. Table 5 presents that adopting LLaMA (Touvron et al., 2023) / LLaVA (Liu et al., 2023) can slightly outperform CLIP, and the visual-aware encoding is also crucial in the original InsPix2Pix. However, they still contain a performance gap with our MGIE, which indicates that merely replacing the instruction encoder is not enough for their limitation. We further consider HIVE (Zhang et al., 2023c) and InsEdit (Wang et al., 2023a) for the additional baselines. HIVE collects human preference and enhances InsPix2Pix via reward feedback learning. InsEdit depends on an external segmentation model to provide the target mask and performs inpainting as the editing result. The results demonstrate that MGIE consistently surpasses HIVE without extra human feedback, which is more data-efficient for training. InsEdit is superior in local editing with its mask-then-inpaint but not in global optimization. The mask should always be the entire photo, and the inpainting is not capable of adjusting the brightness or saturation. In contrast, through learning with the derivation of the MLLM, our MGIE performs robustly in both.

increase the brightness

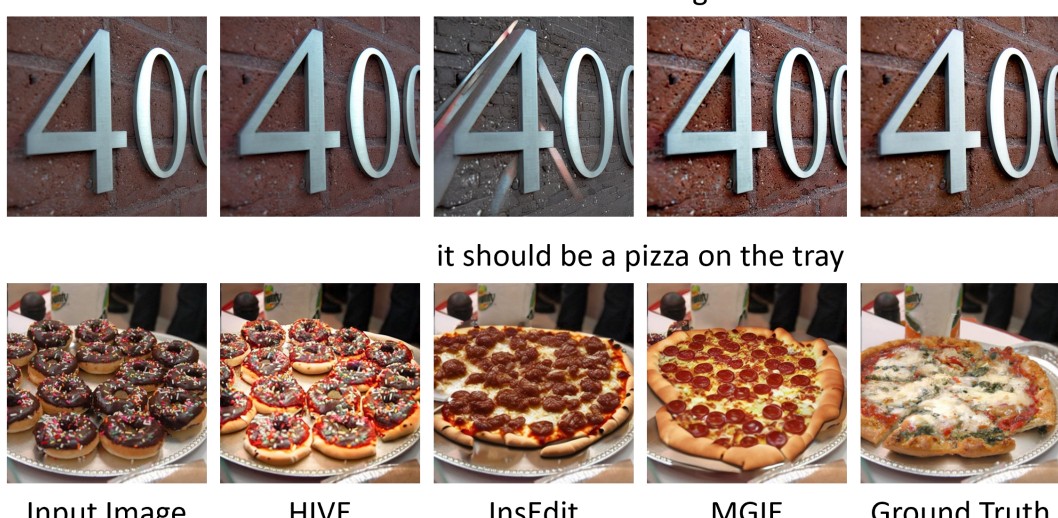

it should be a pizza on the tray

| Input Image | HIVE | InsEdit | MGIE | Ground Truth |

**Does Larger LM Help?** Our MGIE leverages LLMs/MLLMs to enhance instruction-based image editing. We investigate that if stronger LMs can bring more improvement. We consider the visual-tuned OPT-6.7B (Zhang et al., 2022) and the larger LLaVA-13B in Table 6. We also adopt LLaMA-13B for LGIE. Even though MGIE-7B has a similar size to MGIE-6.7B, its LLaVA is more powerful than OPT, which leads to an accurate visual imagination for better editing. The 13B obtains further performance gain for both LGIE and MGIE. Fig. 9 plots the CLIP-Score of expressive instructions by different sizes of MGIE. This indicates that the guidance from larger LMs is more alignment with the vision, and thus can benefit image editing more.

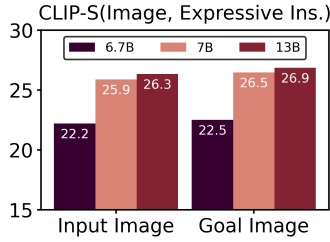
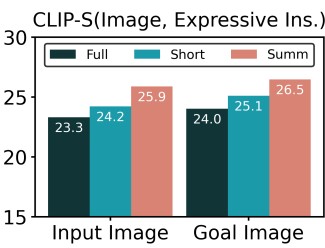
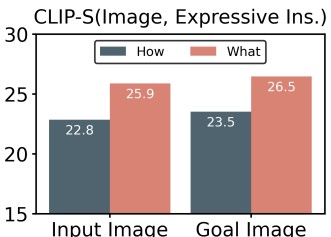

Figure 9: **CLIP-S** across images and expressive instructions by different sizes of MGIE.

Figure 10: **CLIP-S** across images and expressive instructions (full / short / summarized).

Figure 11: **CLIP-S** across images and expressive instructions by the "*how*" or "*what*" prompt.

**Learning with Summarized Expressive Instruction.** By default, MGIE learns with summarized expressive instructions for better performance and inference efficiency. We compare our form to the full description and the one making "*what will this image be like if* `[INS]` *(in short)*" as the prompt. Fig. 10 illustrates that Full is not that aligned with images due to its irrelevant narrations (*e.g.*, "*filled with people enjoying the waterfront*"). Although Short can derive brief statements (21.1 tokens), our Summ (22.7 tokens) is still more aligned with input or goal images. In the qualitative aspect, Short's "*create a dangerous element*" is not explicit for "*add a shark*". Short even merely captions the photo but without "*in Seattle*". In contrast, our Summ provides concise yet concrete guidance, such as "*a shark swimming on the lake*" or "*iconic Space Needle, urban setting*".

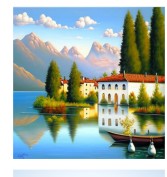

**add a shark**

This would create an unexpected and potentially dangerous element ... The shark would evoke a sense of danger and unease, as sharks are known for being predators ... potentially disrupt the harmony ...

Add a shark to create a dangerous element, disrupting harmony and creating an alarming visual.

The presence would showcase a shark swimming on the lake, with a boat and building nearby.

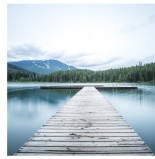

**make it in Seattle**

The image would be transformed into a vibrant and lively scene, showcasing the city's iconic landmarks ... The dock would be filled with people enjoying the waterfront, engaging in various activities ...

This image will be like a scene in Seattle, with a wooden dock and blue lake surrounded by mountains.

The image features iconic landmarks like the Space Needle. The wooden dock would be in an urban setting.

| Input Image | Full | Short | Summ |

Apart from the used "*What*" prompt, we also investigate a "*How*" prompt as "*how to edit this image and* `[ins]`" for expressive instructions. Fig. 11 shows that our "*What*" is more aligned, which can guide image editing with more relevant visual implications, such as "*painted in hues of red, orange, and yellow*" for Autumn or "*famous landmarks as Kremlin*" for Russia. "*How*" miscomprehends the instruction as "*replace the whole garden with a beach*". However, it should only manipulate the end of the stairs yet remain "*the stairway surrounded by lush greenery*".

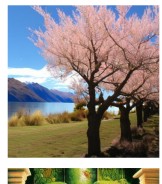

**make it a pine tree**

Change the color of the tree to green. Manually paint green color onto the tree's branches and leaves.

It would depict a serene landscape featuring a large pine tree standing tall next to a body of water.

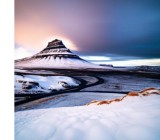

**turn the season into autumn**

Apply color grading techniques to adjust the color temperature and make the image look more autumnal.

The snow-covered mountain is painted in hues of red, orange, and yellow, as the leaves change colors.

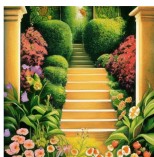

**have the stairs lead to a beach**

Use the clone brush tool to replace the whole garden scene with a beach scene in the image.

As the viewer follows the stone steps, they would see a beach. The stairway is surrounded by lush greenery.

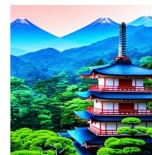

**make the trip to russia**

Fill in any gaps or missing details. Take the image to Russia and explore its rich history and natural beauty.

The image could show the famous landmarks such as the Kremlin, Red Square, or St. Basil's Cathedral.

| Input Image | How | What | Input Image | How | What |

**How Many Visual Tokens do We Need?** Our editing head projects the guidance modality from the MLLM to the diffusion model. We follow GILL (Koh et al., 2023) and apply $N{=}8$ visual tokens by default. Here we investigate the effectiveness of different numbers of `[IMG]`. The results indicate that less `[IMG]` makes the extracted visual imagination insufficient for effective guidance, resulting

in a significant performance drop. While more `[IMG]` can bring further enhancements, we also find that the performance gets similar when using more than 4 `[IMG]`.

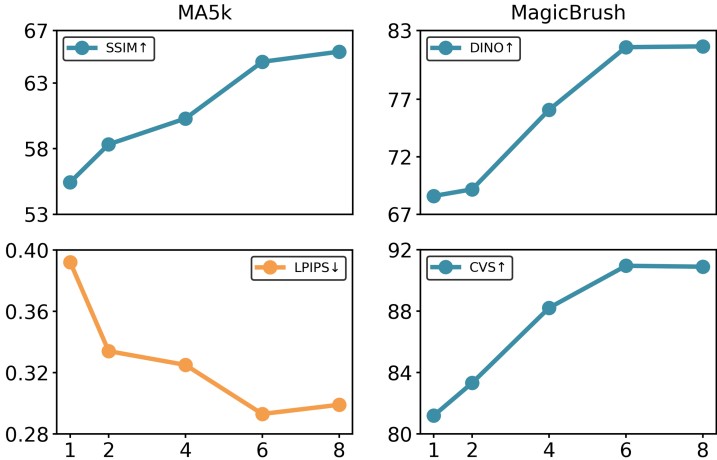

**Qualitative Results of Different $\alpha_{\mathcal{V}}$.** MGIE adopts the weight $\alpha_{\mathcal{V}}$ to adjust the level of editing. A higher $\alpha_{\mathcal{V}}$ makes the editing result more similar to the input, while a lower $\alpha_{\mathcal{V}}$ leads to more editing applied onto the image. Hence we can control the extent of visual transformation for both local (*e.g.,* the color of cherries) and global editing (*e.g.,* the style of the painting).

make the cherry ripe purple

the forest path to a beach

much more abstract

| Input Image | $\alpha_{\mathcal{V}}$ = 2.2 | 1.8 | 1.4 | 1.0 |

**Comparison to Description-based Baselines.** In addition to instruction-based baselines, we also consider description-based editing models. We leverage GIT (Wang et al., 2022) to caption the input image as its input description and ChatGPT to merge the edit instruction as the goal description via the prompt "*Combine two sentences A:* `[description]` *and B:* `[instruction]` *into a single sentence. The output should be at most similar to sentence A*". For instance, "*a girl is walking at the beach*" and "*give her a hat*" will be transformed into "*a girl with a hat is walking at the beach*". For

MagicBrush, we directly apply their released descriptions instead. Text2LIVE (Bar-Tal et al., 2022) and Null-Inv (Mokady et al., 2022) only yield feasible results on the traditional L1 distance but are obviously inferior to our MGIE on semantic-level evaluations (*e.g.,* lower CVS), which supports that they cannot present concrete editing results and carry out goal descriptions well. On the other hand, both count on inference optimization (CLIP alignment and DDIM inversion), which takes more than 200 seconds (*vs.* ours 9.2 seconds) for each editing task.

| Method | EVR | | | GIER | | | MA5k | | | MagicBrush | | | |
|---|---|---|---|---|---|---|---|---|---|---|---|---|---|
| | L1↓ | DINO↑ | CVS↑ | L1↓ | SSIM↑ | CVS↑ | L1↓ | SSIM↑ | LPIPS↓ | L1↓ | DINO↑ | CVS↑ | CTS↑ |
| Text2LIVE | 0.169 | 66.19 | 78.22 | **0.126** | **58.32** | 79.32 | 0.165 | 57.62 | 0.342 | **0.071** | **83.35** | 89.71 | 23.59 |
| Null-Inv | 0.174 | 69.24 | 78.35 | 0.149 | 58.24 | 82.33 | 0.179 | 61.36 | 0.335 | 0.073 | 81.72 | 87.24 | 27.62 |
| InsPix2Pix | 0.189 | 67.82 | 81.38 | 0.144 | 57.51 | 86.63 | 0.176 | 58.92 | 0.359 | 0.101 | 71.46 | 85.22 | 29.34 |
| MGIE | **0.163** | **71.49** | **81.73** | 0.135 | 59.24 | **88.59** | **0.133** | **66.25** | **0.298** | 0.082 | 82.22 | **91.14** | **30.40** |

**Evaluating Image Editing via FID.** As ground-truth goal images are available, we also calculate the Fréchet inception distance (FID) for editing results under the zero-shot or fine-tuned evaluation. However, the differences are all pretty limited. Since most editing results still resemble the original input images, it is difficult for FID to determine their authenticity. These results indicate that FID is insufficient to compare the quality of image editing.

| Method | Zero-shot | | | | Fine-tuned | | | |
|---|---|---|---|---|---|---|---|---|
| | EVR | GIER | MA5k | MagicBrush | EVR | GIER | MA5k | MagicBrush |
| InsPix2Pix | **6.19** | **5.61** | 5.91 | 5.69 | **5.31** | **5.31** | **5.30** | 5.64 |
| LGIE | 6.67 | 5.69 | 5.80 | **5.31** | 5.32 | 5.42 | 5.59 | 5.48 |
| MGIE | 6.45 | 5.64 | **5.48** | 5.61 | 5.53 | 5.59 | 5.41 | **5.42** |

**Part-of-Speech Distribution.** We investigate part-of-speech (POS) distributions[5] of input instructions and our derived expressive instructions. In general, input instructions involve more nouns but fewer adjectives. In contrast, our expressive instructions can portray concrete edited scenes in detail via more adjectives. The original instructions are also dominated by verbs, which are challenging to perceive. The derivation helps them to be more understandable as adverbs. Moreover, we effectively decrease the number of ambiguous pronouns. More than 68% pronouns (only 13% in our expressive instructions) are unresolvable in input instructions[6], where the model can not have explicit goals.

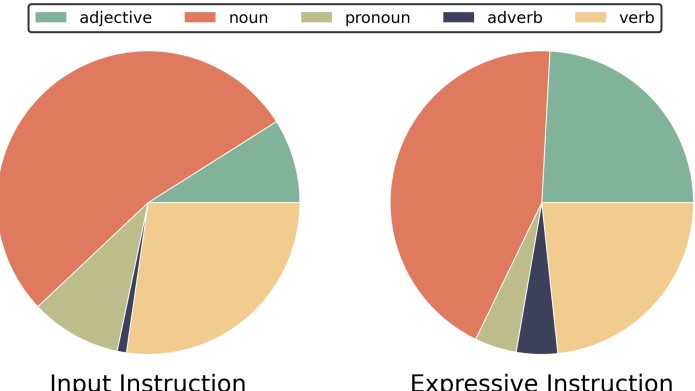

Input Instruction      Expressive Instruction

**Unseen Editing Operation.** Since there is no removal or photo optimization in IPr2Pr, InsPix2Pix has failed due to the shortage of training examples. Our MGIE is able to handle such editing via the visual-aware derivation of MLLM. We can accurately remove "*the boy in red shirt*" or "*lighten out the yellow tone*", which demonstrates better generalizability for unseen operations. More qualitative comparisons can be found on our project website[4].

---

[5]We adopt flairNLP (`https://github.com/flairNLP/flair`) as the part-of-speech tagger.
[6]We apply AllenNLP (`https://github.com/allenai/allennlp`) for coreference resolution.

remove text

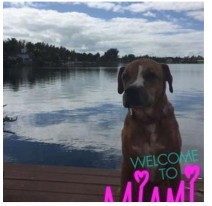 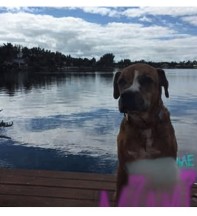 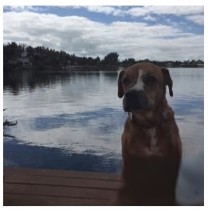

remove boy with red shirt from picture

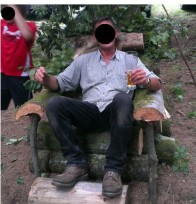 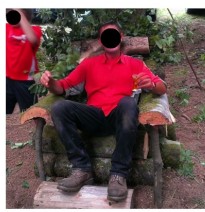 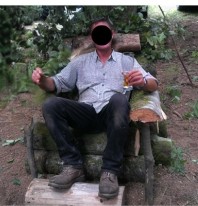

remove hot air balloon

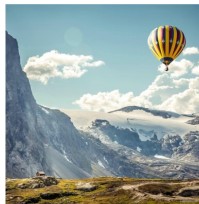 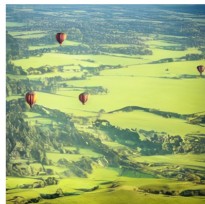 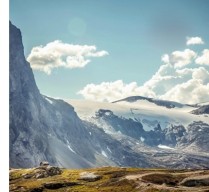

need to clarified, more focus

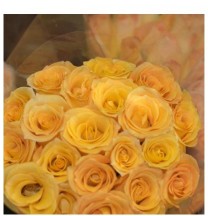 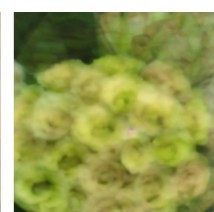 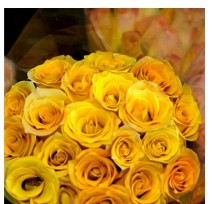

please reduce the brightness of the image

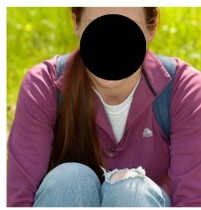 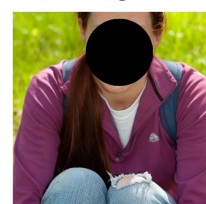 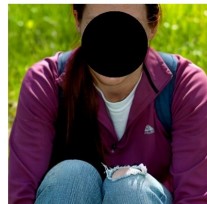

lighten out yellow tone

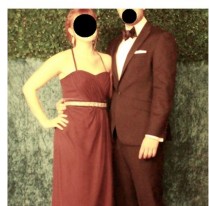 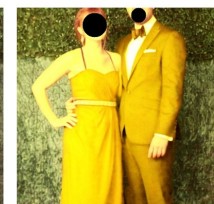 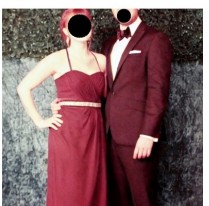

Input Image      InsPix2Pix      MGIE

**Ablation Study of Training Loss.** There are two training losses, instruction loss ($\mathcal{L}_{ins}$) and editing loss ($\mathcal{L}_{edit}$), in our MGIE. $\mathcal{L}_{edit}$ is necessary for training to produce the editing result. Without $\mathcal{L}_{ins}$, it will derive full but lengthy guidance to lead $\mathcal{L}_{edit}$. However, both LGIE and MGIE drop significantly; LGIE even performs worse than the baseline. This underscores the prominence of learning concise expressive instructions, which offer succinct and relevant guidance. Besides, lengthy instructions via the MLLM will incur additional overhead (29.4 *vs.* ours 9.2), resulting in an inefficient inference.

| Method Setting | | MA5k | | MagicBrush | | |
|---|---|---|---|---|---|---|
| | | SSIM↑ | LPIPS↓ | DINO↑ | CVS↑ | CTS↑ |
| InsPix2Pix | | 58.92 | 0.359 | 71.46 | 85.22 | 29.34 |
| LGIE | - $\mathcal{L}_{ins}$ | 57.59 | 0.386 | 70.79 | 83.21 | 28.66 |
| | + $\mathcal{L}_{ins}$ | **64.60** | **0.327** | **80.90** | **88.87** | **30.10** |
| MGIE | - $\mathcal{L}_{ins}$ | 58.18 | 0.365 | 71.50 | 85.19 | 29.11 |
| | + $\mathcal{L}_{ins}$ | **66.25** | **0.298** | **82.22** | **91.14** | **30.40** |

**Adding New Object.** MGIE also supports adding new objects that are not present in the input and placing them in reasonable positions. For instance, the "*hat*" is put on the girl's head, and the "*river*" is added along with the grass. More surprisingly, the appended "*fireworks*" further makes the beach colorful, which drives the night scene coherent and visually appealing.

have a birthday cake on table     give her a hat

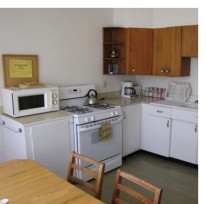 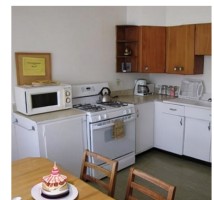 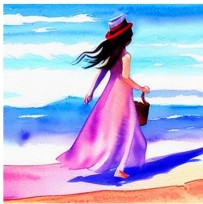

put a river nearby     add fireworks into the sky

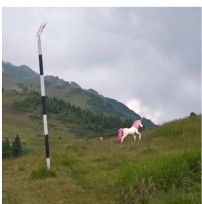 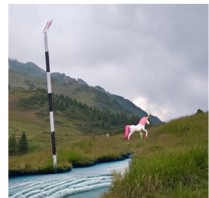 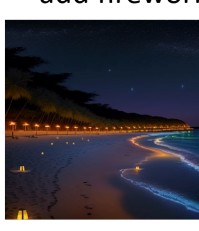 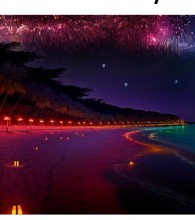

Input Image  MGIE   Input Image  MGIE

**Transferring Image Texture/Color/Emotion.** We attempt transferring visual patterns of images, also controlled through human instructions. For texture, we follow CLVA (Fu et al., 2022) and adopt the style prompt "*make the whole image as texture* `[ins]`". InsPix2Pix can only do limited transfer, but MGIE shows clear visual attributes (*e.g.,* "*orange*" or "*pinkish*") as well as the complex "*colorful circular round*". We perform fine-grained color manipulation, including "*glasses frame*" or "*hair*". However, the baseline even alters the whole color. For global colorization (Chang et al., 2023), both InsPix2Pix and our MGIE cannot present appealing results, which indicates the need for fine-tuning. Transferring the emotion is more challenging as the model has to perceive the latent semantics. We are able to illustrate the visual concept of "*bright day*" or "*chaotic and confused*" as the beach in the early morning or the gloomy street at night. MGIE can also transform from the cozy snowy day into suspenseful and thrilling through "*nightmare and scared*". Although exhibiting promising potential, it still requests more profound texture/emotion perception for each specific goal. We leave them as future research for creative visual editing (Weng et al., 2023).

hexagonal, orange, blue, smooth white

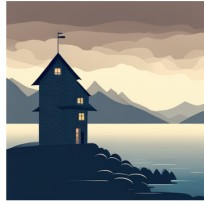 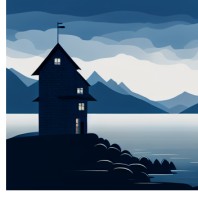 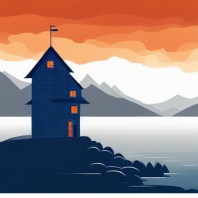

pinkish, interlaced, cloth, like pillow cover

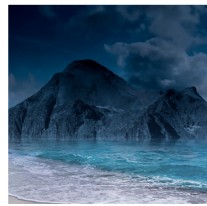 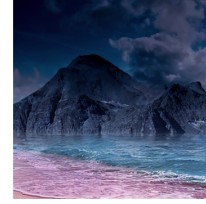 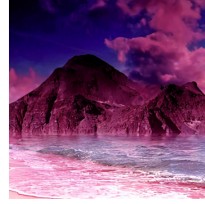

colorful smooth pretty circular round

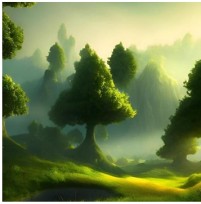 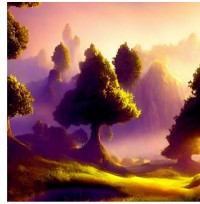 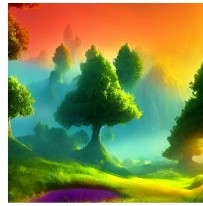

Input Image          InsPix2Pix          MGIE

*color/emotion results on the next page*

make the frame red

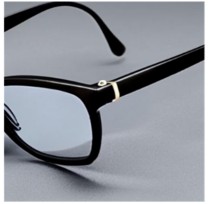 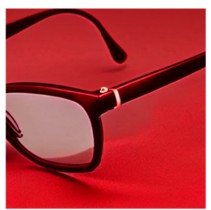 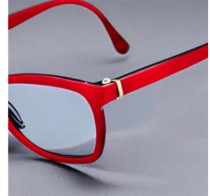

change the hair to green color

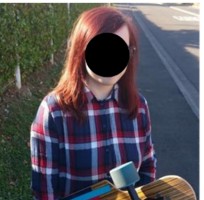 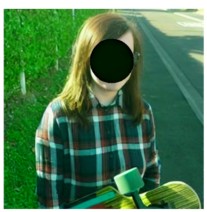 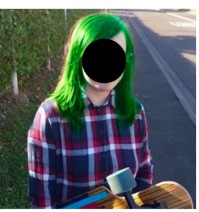

as a colorful image

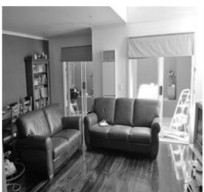 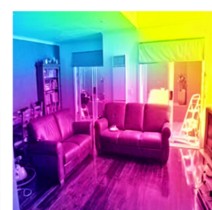 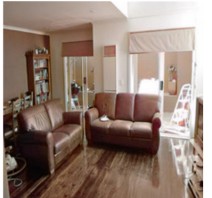

feel chaotic and confused due to the tone

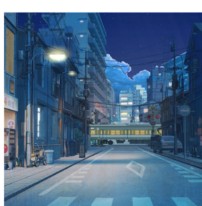 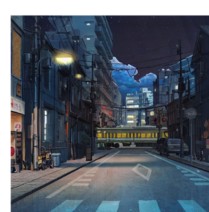 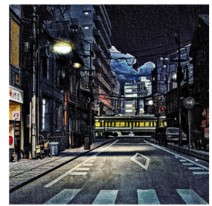

charmed by the beautiful bright day

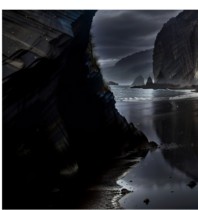 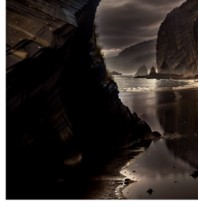 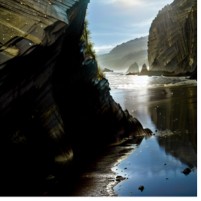

out of nightmare, utterly scared and shaken

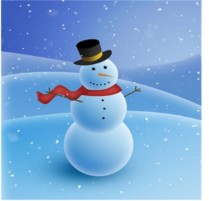 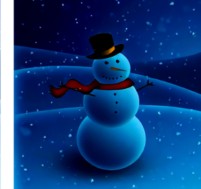 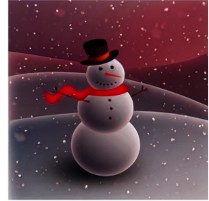

Input Image      InsPix2Pix      MGIE

## B    DETAILED EXPERIMENTAL SETUP

**Edit Head to Joint the MLLM and the Diffusion Model.**    These appended visual tokens [IMG] are treated as the latent imagination of the editing goal from the MLLM but in the language modality. Inspired by GILL (Koh et al., 2023), we consider an edit head $\mathcal{T}$ to transform them into actual visual guidance. $\mathcal{T}$ is a lightweight 4-layer Transformer, which takes word embeddings $e$ and hidden states $h$ of [IMG] as the input and generates the visual imagination $\{u_1, ..., u_L\}$, conditioned on learnable query embeddings $\{q_1, ..., q_L\}$. As our diffusion model is inherited from StableDiffusion (Rombach et al., 2022), we apply the same $L = 77$, and the dimension of $u$ is 768.

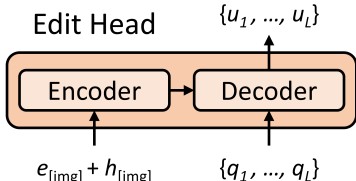

**Editing Loss of the Diffusion Model.**    Our diffusion model is built upon latent diffusion $\mathcal{F}$ (Rombach et al., 2022), which operates the latent space of the variational autoencoder (VAE). For the goal image $\mathcal{O}$, the diffusion process keeps adding noises to the encoded $o = \mathrm{Enc}_{\mathrm{VAE}}(\mathcal{O})$ and produces a noisy latent $z_t$. Our target is to learn the UNet $\epsilon_\theta$ that predicts the added noise according to the input image $v = \mathrm{Enc}_{\mathrm{VAE}}(\mathcal{V})$ and the visual imagination $\{u\}$ from the MLLM. The learning objective is:

$$\mathcal{L}_{\mathrm{edit}} = \mathbb{E}_{o,v,\{u\},\epsilon \sim \mathcal{N}(0,1),t} \left[ ||\epsilon - \epsilon_\theta(z_t, t, v, \{u\})||_2^2 \right].$$

Following InsPix2Pix (Brooks et al., 2023), we leverage the classifier-free guidance (Ho & Salimans, 2021), which combines both conditional and unconditional (a fixed null value $\varnothing$) denoising. During inference, we let the score estimation $s_\theta$ extrapolate toward the conditional yet keep away from the unconditional guidance. Since there are two conditionings ($v$ for image and $\{u\}$ for instruction), our modified $s_\theta$ should be:

$$\begin{aligned} s_\theta(z_t, v, \{u\}) = {} & s_\theta(z_t, \varnothing, \varnothing) \\ & + \alpha_\mathcal{V} \cdot (s_\theta(z_t, v, \varnothing) - s_\theta(z_t, \varnothing, \varnothing)) \\ & + \alpha_\mathcal{X} \cdot (s_\theta(z_t, v, \{u\}) - s_\theta(z_t, v, \varnothing)), \end{aligned}$$

where we randomly set $v = \varnothing$, $\{u\} = \varnothing$, or both $= \varnothing$ for 5% of data during training. $\alpha_\mathcal{V}$ and $\alpha_\mathcal{X}$ are guidance scales to control the trade-off between input image similarity and instruction alignment. By default, we use $\alpha_\mathcal{V} = 1.5$ and $\alpha_\mathcal{X} = 7.5$.

**Training Cost.**    Our MGIE training requires 26 epochs to converge, and InsPix2Pix has 20 epochs (from their released checkpoint). Both MGIE and InsPix2Pix take a similar 1.6 hours per epoch on our node (8 NVIDIA A100 GPUs), where the overall training can be done in two days.

**Human Evaluation.**    We sample 100 examples (25 for each dataset) to conduct our human evaluation. Each task is assigned 3 annotators, who rank across baselines and our MGIE, to avoid potential bias. We require workers to have a 97% approval rate and over 500 approved tasks to ensure quality. The worker is awarded $5 for each task (5 examples) and takes 21 minutes on average to complete.

## C    ETHICS DISCUSSION AND LIMITATION

In this paper, we leverage multimodal large language models (MLLMs) with the diffusion model to enhance instruction-based image editing. Even though our work benefits creative visual applications, there are still limitations that should be taken into consideration when interpreting the results. Since our MGIE is built upon pre-trained foundation models, it is possible to inherit bias from LLaVA and StableDiffusion. To mitigate this issue, we make the derived expressive instruction concise through summarization and update the MLLM together with the diffusion model. This end-to-end learning can also reduce the potential harmfulness since the hallucination from the LM will not be expressed over the editing. We can incorporate the safety checker (Rombach et al., 2022) to filter out offensive results during post-processing as the final line of defense. From the perspective of editing, there are

some challenging cases. Compositional command is hard to accomplish in a single step. Our MGIE can successfully remove the left sign but not the subsequent manipulation. In addition, the ability of language grounding (*e.g.,* only the potato should be replaced), as well as numerical perception (*e.g.,* just add to one cupcake), can be improved for more accurate targeting. We leave these directions as future research to achieve more practical and powerful instruction-based image editing.