# OpenReview forum: "Guiding Instruction-based Image Editing via Multimodal Large Language Models"
_ICLR.cc/2024/Conference — ICLR 2024 spotlight_

### Official Review · Reviewer_TXUn · 2023-10-22

**Soundness:** 4 excellent
**Presentation:** 3 good
**Contribution:** 3 good
**Rating:** 8
**Confidence:** 4

**Summary:**

The paper introduces an image editing method guided by MLLM. This approach can learn from expressive instructions and offer explicit guidance. Comprehensive experiments demonstrate the method's effectiveness.

**Strengths:**

The paper is clearly written.

The introduced technique.is novel and interesting.

The experiments are sufficiently conducted.

The presented results seems promising.

**Weaknesses:**

While certain aspects of the work might appear less novel, its practical effectiveness compensates for this.

There are typographical errors. Specifically, "Methods" in Tables 1 and 2 should be placed at the top.

**Questions:**

I am curious about the method's potential in handling tasks like:

1. Transferring image texture. Language-Driven Artistic Style Transfer, ECCV 2022.

2. Modifying an object's color. L-CAD: Language-based Colorization with Any-level Descriptions using Diffusion Priors, NeurIPS 2023.

3. Interpreting user-provided emotions. Affective Image Filter: Reflecting Emotions from Text to Images, ICCV 2023.

Including comparisons or referencing these studies could further enrich the paper's depth and context.

---

> ### Author Response · Authors · 2023-11-18
>
> We thank reviewer TXUn for acknowledging our practical effectiveness.
>
> > **Q) typographical errors**
>
> Thanks for the debugging! We have updated all Tables in our revision.
>
> > **Q) transferring texture/color/emotion**
>
> We attempt to perform such transfers in Sec. A (Transferring Image Texture/Color/Emotion). For texture, we follow CLVA [1] and adopt the style prompt "*make the whole image as texture [ins]*". InsPix2Pix can only do limited transfer, but MGIE shows clear visual attributes (e.g., "*orange*" or "*pinkish*") as well as the complex "*colorful circular round*". We perform fine-grained color manipulation, including "*glasses frame*" or "*hair*". However, the baseline even alters the whole color. For global colorization [2], both InsPix2Pix and our MGIE cannot present appealing results, which indicates the need for fine-tuning. Transferring the emotion is more challenging as the model has to perceive the latent semantics. We are able to illustrate the visual concept of "*bright day*" or "*chaotic and confused*" as the beach in the early morning or the gloomy street at night. MGIE can also transform from the cozy snowy day into suspenseful and thrilling through "*nightmare and scared*". Although exhibiting promising potential, it still requests more profound texture/emotion perception for each specific goal. We leave them as future research for creative visual editing [3].
>
> + [1] (ECCV'22) Language-Driven Artistic Style Transfer
> + [2] (NeurIPS'23) L-CAD: Language-based Colorization with Any-level Descriptions using Diffusion Priors
> + [3] (ICCV'23) Affective Image Filter: Reflecting Emotions from Text to Images

---

> > ### Comment · Reviewer_TXUn · 2023-11-21
> >
> > I extend my thanks to the authors for their diligent efforts. Based on the merits of their work, I am inclined to maintain my rating in favor of acceptance.

---

### Official Review · Reviewer_RXEx · 2023-10-28

**Soundness:** 3 good
**Presentation:** 3 good
**Contribution:** 3 good
**Rating:** 8
**Confidence:** 4

**Summary:**

This paper proposes a new instruction-based image editing approach. The Multimodal large language models (MLLMs) are incorporated into the editing, formulating MLLM-Guided Image Editing (MGIE). Different editing operations are utilized for testing, including Photoshop-style modification, global photo optimization, and local editing.

**Strengths:**

The proposed method can have effective performance on the chosen datasets of evaluation. Especially, a lot of evaluations are conducted with subjective assessment.

**Weaknesses:**

1.	The performance is limited by the dataset utilized for training, especially for learning effective instruction and editing performance. Will this approach complete the editing operation which is not appeared in the training data?

2.	The template of this paper is wrong, it is still the template of ICLR2023.

3.	There is no ablation analysis for the loss functions and their alternatives.

4.	The editing effects are not ideal, for example, the woman in the background of Fig. 1 is removed, while there are still some residuary artifacts in the background.

**Questions:**

1.	How to balance the loss weights of $\mathcal{L}_{ins}$ and $\mathcal{L}_{edit}$ in Eq. 6?

2.	Can this approach add new objects into the image?

3.	What are the details of the user study’s participants? Like their ages, educations, and the time for taking part in the user study.

**Details Of Ethics Concerns:**

The generated image may be harmful to the protection of copyright.

---

> ### Author Response · Authors · 2023-11-19
>
> We thank reviewer RXEx for enjoying our proposed method.
>
> > **Q) unseen editing operation**
>
> We first want to clarify that MGIE is not limited by the training dataset. In IPr2Pr, there is no removal or photo optimization. We have appended such cases in Sec. A (Unseen Editing Operation). InsPix2Pix has failed due to the shortage of training examples. In contrast, our MGIE is able to handle such editing via the visual-aware derivation of MLLM. We can accurately remove "*the boy in the red shirt*" or "*lighten out the yellow tone*", which demonstrates better generalizability for unseen operations. More qualitative comparisons can be found on our project website (https://mllm-ie.github.io).
>
> > **Q) ICLR template**
>
> Thanks for the notification! We have updated the ICLR 2024 template in our revision.
>
> > **Q) ablation study of training loss**
>
> We have added the study in Sec. A (Ablation Study of Training Loss). There are two training losses, instruction loss $L_\text{ins}$ and editing loss $L_\text{edit}$, in our MGIE. $L_\text{edit}$ is necessary for training to produce the editing result. Without $L_\text{ins}$, it will derive full but lengthy guidance to lead $L_\text{edit}$. However, both LGIE and MGIE drop significantly; LGIE even performs worse than the baseline. This underscores the prominence of learning concise expressive instructions, which offer succinct and relevant guidance. Besides, lengthy instructions via the MLLM will incur additional overhead (29.4 vs. ours 9.2), resulting in an inefficient inference.
>
> | Method | Setting | Ma5k | | MagicBrush | | |
> | :-- | :-- | :-- | :-- | :-- | :-- | :-- |
> | | | **SSIM↑** | **LPIPS↑** | **DINO↓** | **CVS↑** | **CTS↑**
> | InsPix2Pix | | 58.92 | 0.359 | 71.46 | 85.22 | 29.34 |
> | LGIE | - $L_\text{ins}$ | 57.59 | 0.386 | 70.79 | 83.21 | 28.66 |
> | | + $L_\text{ins}$ | **64.60** | **0.327** | **80.90** | **88.87** | **30.10** |
> | MGIE | - $L_\text{ins}$ | 58.18 | 0.365 | 71.50 | 85.19 | 29.11 |
> | | + $L_\text{ins}$ | **66.25** | **0.298** | **82.22** | **91.14** | **30.40** |
>
> > **Q) editing effects are not ideal**
>
> We also observe these artifacts. Though the editing results are not perfect, MGIE can effectively remove the target, compared to the failed InsPix2Pix. Perhaps a step-by-step pipeline (e.g., removal followed by inpainting) could deal with such issues.
>
> > **Q) balance between $L_\text{ins}$ and $L_\text{edit}$**
>
> As Eq. 6, the overall training loss is $L_\text{ins} + 0.5 \cdot L_\text{edit}$. We have tried different weights (e.g., 1.0), but the performances are all similar.
>
> > **Q) add new objects**
>
> We have appended such examples in Sec. A (Adding New Object). MGIE also supports adding new objects that are not present in the input and placing them in reasonable positions. For example, the "*hat*" is put on the girl's head, and the "*river*" is added along with the grass. More surprisingly, the appended "*fireworks*" further makes the beach colorful, which drives the night scene coherent and visually appealing.
>
> > **Q) details of user study**
>
> We have updated the details in Sec. B (Human Evaluation). We sample 100 examples (25 for each dataset) to conduct our human evaluation. Each task is assigned 3 annotators, who rank across baselines and our MGIE, to avoid potential bias. We require workers to have a 97% approval rate and over 500 approved tasks to ensure quality. The worker is awarded $5 for each task (5 examples) and takes 21 minutes on average to complete.

---

> > ### Author Response · Authors · 2023-11-21
> >
> > Dear Reviewer RXEx, we are thankful for your review. As the rebuttal deadline is coming to an end, please let us know if your concerns are well addressed. We are happy to provide further clarification.

---

### Official Review · Reviewer_fMJn · 2023-10-30

**Soundness:** 3 good
**Presentation:** 3 good
**Contribution:** 2 fair
**Rating:** 6
**Confidence:** 3

**Summary:**

This submission proposes an approach towards improving the visual output quality of instruction-based image editing. Authors put forward the hypothesis that natural text-based edit instructions, provided by humans, are oft terse and this hinders contemporary learning-based models from successfully capturing and dependably following intended image-edit meanings.

The crux of the proposed strategy involves learning to map terse input text, representing image edit-instructions, to more expressive (evidently more verbose) output text, in order to provide more explicit image-editing guidance. The work explores the efficacy of leveraging cross-modal capabilities, contemporary language-models, and sensitivity to visual inputs in order to positively influence edit-instruction quality. This multi-modal component is evidenced to be important for text-edit quality and resulting downstream edited images. Quantitative and qualitative evaluation of the proposed strategy, across various metrics and human rankings, are reported in comparison with recent work and a relevant baseline, across four benchmark datasets.

**Strengths:**

The problems being addressed here are real and are important -- principled solutions and progress towards techniques capable of realising consistently high quality and complex (intended) image edits, that are also of low cost in terms of required human effort, will be of high value to the community and will additionally result in widely applicable and practical end-user benefits.

The technical components of the work, the manner in which the various components are concatenated together, appears reasonably straightforward. Explanation of the guided image editing strategy is aided by basic yet clear schematics to aid understanding (Fig. 2, specifically).

The investigation can be considered reasonably thorough with the inclusion of experimental work covering (i) method efficacy (quant. & qual.), (ii) vision & language hyper-parameter sensitivity, (iii) ablating instruction generation components, (iv) compute. Reported qualitative results are quite compelling and show some good perceptual improvements over baselines. Further results (public anon-web page, supp. materials) are appreciated and provide some aid for apprehensions over method robustness, reliability.

Writing is of a reasonable standard in general. I enjoyed reading the paper.

**Weaknesses:**

Leveraging multi-modal language-models provides an intuitively promising avenue, when striving to follow image-manipulation intentions that are defind by only terse human text instruction. Good evidence is provided that the presented strategy goes some way to tackling the observed short-comings however technical contributions (size, sufficiency) can be regarded as moderate. The overarching system makes use of an array of pre-existing components however the particular implementation that facilitates component concatenations can be regarded as somewhat novel.

For this venue, an important factor that would strengthen the submission might look to provide additional understanding, discussion on the long-term sustainability of the proposed style of approach. The requirement for intermediary 'on-the-fly' remappings, reconfigurations of text input cannot be regarded as a very elegant or pleasing solution. I explicitly note that this is not grounds for rejection, my point is rather that further thinking, discussion on the fundamental gaps that prevent base models from correctly realising terse (yet human-parseable) instruction would likely prove elucidating and of high value. An empirical avenue here might involve investigating pre- and post-hoc parts-of-speech (POS) distributions, or other statistical evaluation of the edit-text distributions. Is the long-term goal, alternatively, to fashion single models, capable to understand natural (terse) human instruction? The point touches on well-understood (dis-)advantages of modular-component systems c.f. end-to-end.

I would be keen for authors to discuss these points and I am open to modifying my score.

Minor suggestions:

1. Authors may wish to update to an ICLR24 template (c.f. ICLR23)

2. On pp.2: suggest explicitly expand acronym 'IPr2Pr' on first use (presumably Instruction Prompt-2-Prompt ?)

3. Precision of some phrasing could be tightened, towards aiding reading understanding. See example suggestions in following section.

4. The (useful!) model schematic image, found between Figure 3 and Figure 4, is unnumbered. Is this by design ? Suggest 'Baselines' paragraph might serve as suitable caption content.

**Questions:**

* Can the authors commit to a full source code release? The submission opted to stay silent on this issue. Code release would be of clear benefit to the community aiding reproducibility, public probing of method robustness, consistency, range of model abilities. This will undoubtedly increase the value of the contributions. If impossible; web page might be extended to allow public inference-time testing.

* The property of 'elaborate descriptions' is posed as something one wishes to evade in natural commands and yet 'explicit yet detailed guidance' is conversely noted a sought after notion. Do the authors suggest that the former pertains largely to image space, in a similar fashion to e.g. regional masks? Concrete examples of the considered 'elaborate descriptions' may help to clarify this point.

* How is the phrase 'reasonable image editing' to be defined?

* Minor: Small icons (flame, blue cube) tagged to components in the system schematic (Fig.2) presumably represent learnable and frozen model components, respectively. Suggest to make this key explicit.

**Details Of Ethics Concerns:**

Authors provide a brief yet reasonable discussion on ethical issues and limitations in their supp. materials. I have no noteworthy outstanding ethical concerns.

---

> ### Author Response · Authors · 2023-11-19
>
> We thank reviewer fMJn for appreciating the conducted results and our paper writing.
>
> > **Q) technical contribution**
>
> We first want to highlight that our deriving human instructions into informative descriptions is a brand-new attempt, which yields surprising enhancements. Though our MGIE is a joint framework with the MLLM and the diffusion model, we want to emphasize that our contribution includes:
> + an end-to-end pipeline to learn/incorporate explicit yet concise guidance;
> + an investigation on how to utilize expressive instructions (Table 3);
> + a comprehensive automatic/human study covering diverse editing aspects.
>
> All of these are valuable contributions to controllable image editing research.
>
> > **Q) intermediary on-the-fly remappings**
>
> We believe that deriving expressive instructions is crucial for enhancing image editing. In Table 5, we demonstrate that replacing the CLIP encoder with powerful LLMs only results in limited improvement. Since LLMs are trained unidirectionally (decoder-only), the hidden states will lack sufficient/concrete information unless we derive the subsequent narration. On the other hand, expressive instructions can also reveal how the model modifies the input image, aiding in the interpretation of the editing result.
>
> > **Q) part-of-speech distribution**
>
> We investigate part-of-speech (POS) distributions of input instructions and our derived expressive instructions in Sec. A (Part-of-Speech Distribution). In general, input instructions involve more nouns but fewer adjectives. In contrast, our expressive instructions can portray concrete edited scenes in detail via more adjectives. The original instructions are also dominated by verbs, which are challenging to perceive. The derivation helps them to be more understandable as adverbs. Moreover, we effectively decrease the number of ambiguous pronouns. More than 68% of pronouns (only 13% in our expressive instructions) are unresolvable in input instructions, where the model can not have explicit goals.
>
> | Instruction | adjective | noun | pronoun | adverb | verb |
> | :-- | :-- | :-- | :-- | :-- | :-- |
> | Input | 8.9 | 53.1 | 9.9 | 0.9 | 27.2 |
> | Expressive | 24.1 | 43.7 | 4.5 | 4.4 | 23.3 |
>
> > **Q) suggestions**
>
> Thanks for the notification! We have updated the ICLR 2024 template as well as "Instruction Prompt-to-Prompt" for IPr2Pr in our revision. We also add the caption (overview architectures of baselines) to Figure 4.
>
> > **Q) code release**
>
> Yes. We will release the code/checkpoints after the review process. We also attach a copy of our codebase in the appendix.
>
> > **Q) elaborate descriptions**
>
> Sorry for the misunderstanding. Those previous methods (e.g., Text2Live [1] and Null-Inv [2]) require human-provided descriptions, which are labor-intensive and not intuitive, compared to edit instructions. For example, to manipulate an image of "*a girl is walking at the beach*". The user needs to specify "*a girl with a hat is walking at the beach*", instead of the more straightforward "*give her a hat*". In contrast, our MGIE can automatically derive the concrete guidance "*the girl will wear a hat and walk at the beach*", where elaborate descriptions are no longer necessary in this scenario. This can significantly enhance the accessibility of the edit instruction.
> + [1] (ECCV'22)Text2LIVE: Text-Driven Layered Image and Video Editing
> + [2] (CVPR'23) Null-text Inversion for Editing Real Images using Guided Diffusion Models
>
> > **Q) reasonable image editing**
>
> We call it "*reasonable*" since our model can follow the reasoning from the MLLM. MGIE leverages the derived expressive instruction to perform related image manipulation, where the guidance from the MLLM can offer a reasonable editing goal through powerful language modeling. For example, "*make it more healthy*" for a pizza can be perceived as "*need more vegetables*"; "*add fireworks*" will alter not only the sky but also the reflection on the ground. We calculate the CLIP-Score between edited results and their input/expressive instructions. Our MGIE achieves better alignment than the baseline and shows an even higher correlation with expressive instructions. This indicates that we indeed follow the MLLM and benefit from its reasoning capability. Looking ahead, one potential direction is to adopt the recent GPT-4V [1] to evaluate the actual reasonableness.
>
> | Method | CLIP-S | EVR | GIER | MA5k | MagicBrush |
> | :-- | :-- | :-- | :-- | :-- | :-- |
> | InsPix2Pix | input instruction | 21.19 | 20.36 | 20.31 | 22.58 |
> | MGIE | input instruction | 23.03 | 21.74 | 21.89 | 24.09 |
> | | expressive instruction | 28.33 | 28.25 | 27.69 | 28.37 |
>
> + [1] (arXiv'23) The Dawn of LMMs: Preliminary Explorations with GPT-4V(vision)
>
> > **Q) flame/ice icons**
>
> Thanks for the nice suggestion! We have added it to the caption of Figure 2.

---

> > ### Author Response · Authors · 2023-11-21
> >
> > Dear Reviewer fMJn, we are thankful for your review. As the rebuttal deadline is coming to an end, please let us know if your concerns are well addressed. We are happy to provide further clarification.

---

> > > ### Comment · Reviewer_fMJn · 2023-11-21
> > >
> > > I thank the authors for their detailed rebuttal.
> > >
> > > In particular; additional experimental investigations (POS, CLIP-S) and explicit clarifications on terminology ('elaborate descriptions', 'reasonable image editing') are appreciated.
> > >
> > > Manuscript updates with respect to reviewer BZAL's (valid and somewhat shared) concerns are also noted. Updates related to (i) clear acknowledgement of Koh et al. [1], (ii) multiple random seeds, (iii) description-based baselines, all serve to strengthen the paper.
> > >
> > > In summary I consider reviewer comments well addressed and remain positive about this submission. I lean towards acceptance.
> > >
> > >
> > > 1. Koh et al. 'Generating Images with Multimodal Language Models'. NeurIPS 2023

---

### Official Review · Reviewer_BZAL · 2023-10-31

**Soundness:** 3 good
**Presentation:** 2 fair
**Contribution:** 2 fair
**Rating:** 6
**Confidence:** 3

**Summary:**

This paper proposes a novel text-guided image editing method that builds on the instructpix2pix model, which fine-tunes a diffusion model on instruction-image pairs. Authors propose to leverage a multimodal (image + text) large language model to generate more precise and expressive editing instructions, and improve editing ability. The method is trained on the instructpix2pix dataset, and evaluated on 4 datasets, considering different types of edits.

**Strengths:**

Image editing is a timely and challenging task. Authors demonstrate that they are able to successfully carry out a large set of edit types across multiple datasets. Visual results look very promising, and suggest that the proposed changes can yield noticeable gains over instructpix2pix The evaluation is detailed and the method is analysed from a lot of different angles.

The idea of replacing the CLIP encoder with a more expressive vision-language model is sound, as providing more detailed instructions can help guide the diffusion model towards the desired output. Authors have made efforts to go beyond simple replacement of the text encoder and added functionality (summarization, adaptation to visual content) that improve performance.

**Weaknesses:**

My main concern with this work is the limited novelty. The crux of the innovation is the use of a multimodal language model instead of a CLIP model in the instructpix2pix setting. The second main innovation is the introduction of [IMG] tokens, which are processed by a transformer head to generate conditioning embedding for a LDM model. This approach is very strongly inspired from Koh et al. 2023, where they train a language model to generate image tokens, which are then transformed via a transformer architecture, and used as conditioning for stable diffusion based generation. The source of inspiration should be credited more clearly (the main reference in the method section simply mentions using a similar feature extractor architecture).

The presentation of the paper could be improved as well.  The paper is written in a confusing way, and lacks explanation and justifications for design decisions. For example, equation (5) is introduced without any justifications or intuitive explanation, and author do  not explain what they refer to as by score. Similarly, what authors refer to as MLLMs (pre-trained LLMs adapted to take visual inputs as well) is not clearly defined until section 3.1. Multimodal language models can be designed and trained in different ways (e.g. trained with vision-text inputs jointly), and authors should clarify that they refer to a specific type of models. Similarity, the edit head T was not clearly explained, I needed to read the GILL paper (Koh et al.) to understand how these features were generated. Another example is figure 2, which is mentioned at the beginning of chapter 3.2 and shows a MLLM* model without explanation, while this model is only introduced (in a footnote) in a later paragraph.

While the evaluation is detailed with experiments carried out on many datasets, state of the art references are limited. The only pre-existing work that authors compare to is instructpix2pix, while the LGIE baseline is an overly poor baseline, which is expected to perform worse than any model with vision-language mappings (e.g. CLIP) in a lot of settings. It does not make a lot of sense to ask a pure language model to hallucinate detailed descriptions of an unseen image. Evaluations on prompt-based editing methods are available on the magic brush dataset, and could provide additional context. For consistency, instructions could easily be converted to a prompt using an LLM.

**Questions:**

- Image editing performance can be influenced by the random seed. Results in tables 1-2 compared to results reported in Zhang et al (MagicBrush) show that there can be a noticeable difference for some metrics (e.g. 70->74 for DINO score on instructpix2pix). Several of these evaluations show performance scores that are relatively close, have authors investigated how consistent these rankings are across multiple seeds?

- Are ground truth images (post edit) available for all datasets? Why were these specific sets of evaluation metric chosen? Why not e.g. measure image quality using FID?

- Authors compare inference times, but do not mention training times. How much more more expensive is it to train this new model compared to pix2pix? This is relevant as noticeable performance gains can be observed when fine-tuning on a specific instruction/edit types.

---

> ### Author Response · Authors · 2023-11-19
>
> We thank reviewer BZAL for the valuable feedback and underscoring our promising quantitative/qualitative study.
>
> > **Q) novelty of MGIE**
>
> Thank you for pointing this out! Our MGIE is motivated by GILL [1], and we have made the citation clear in our revision. Despite adopting a similar framework, we are the first to edit an existing image via the MLLM+Diffusion pipeline, rather than creating a new but distinct one. We also want to highlight our deriving human instructions into informative descriptions is a novel attempt at controllable image editing. For the experiments, we investigate the best way to utilize our derived expressive guidance and conduct a comprehensive study of diverse editing aspects, all of which are valuable contributions to this research field.
>
> + [1] (NeurIPS'23) Generating Images with Multimodal Language Models
>
> > **Q) presentation of the model and experimental setting**
>
> Thanks for the constructive feedback! We have appended the detailed experimental setup in Sec. B (Edit Head / Editing Loss). For the MLLM, we first want to clarify that our framework is model-agnostic and compatible with various architectures (as long as they can take visual inputs and provide responses through language modeling). As we are using LLaVA [1] in our implementation, we have made it clearer in Eq. 1. We also added the explanatory footnote of MLLM* to the same page as Figure 2.
> + [1] (NeurIPS'23) Visual Instruction Tuning
>
> > **Q) description-based baselines**
>
> We have involved the results of description-based baselines (TEXT2Live [1] and Null Text Inversion [2]) in Sec. A (Comparison to Description-based Baselines).  We leverage GIT [3] to caption the input image as its input description and ChatGPT to merge the edit instruction as the goal description via the prompt "*Combine two sentences A: [description] and B: [instruction] into a single sentence. The output should be at most similar to sentence A*". For instance, "*a girl is walking at the beach*" and "*give her a hat*" will be merged into "*a girl with a hat is walking at the beach*". For MagicBrush, we directly apply their released descriptions instead. Text2LIVE and Null-Inv only yield feasible results on the traditional L1 distance but are obviously inferior to our MGIE on semantic-level evaluations (e.g., lower CVS), which supports that they cannot present concrete editing results and carry out goal descriptions well. On the other hand, both count on inference optimization (CLIP alignment and DDIM inversion), which takes more than 200 seconds (vs. ours 9.2 seconds) for each editing task.
>
> | Method | EVR | | | GIER | | | MA5k | | | MagicBrush | | | |
> | :-- | :-- | :-- | :-- | :-- | :-- | :-- | :-- | :-- | :-- | :-- | :-- | :-- | :-- |
> | | **L1↓** | **DINO↑** | **CVS↑** | **L1↓** | **SSIM↑** | **CVS↑** | **L1↓** | **SSIM↑** | **LPIPS↓** | **L1↓** | **DINO↑** | **CVS↑** | **CTS↑** |
> | Text2LIVE | 0.169 | 66.19 | 78.22 | **0.126** | 58.32 | 79.32 | 0.165 | 57.62 | 0.342 | **0.071** | **83.35** | 89.71 | 23.59 |
> | Null-Inv | 0.174 | 69.24 | 78.35 | 0.149 | 58.24 | 82.33 | 0.179 | 61.36 | 0.335 | 0.073 | 81.72 | 87.24 | 27.62 |
> | InsPix2Pix | 0.189 | 67.82 | 81.38 | 0.144 | 57.51 | 86.63 | 0.176 | 58.92 | 0.359 | 0.101 | 71.46 | 85.22 | 29.34 |
> | MGIE | **0.163** | **71.49** | **81.73** | 0.135 | **59.24** | **88.59** | **0.133** | **66.25** | **0.298** | 0.082 | 82.22 | **91.14** | **30.40** |
>
> + [1] (ECCV'22) Text2LIVE: Text-Driven Layered Image and Video Editing
> + [2] (CVPR'23) Null-text Inversion for Editing Real Images using Guided Diffusion Models
> + [3] (TMLR'22) GIT: A Generative Image-to-text Transformer for Vision and Language

---

> ### Author Response · Authors · 2023-11-19
>
> > **Q) multiple random seeds**
>
> Thanks for the nice suggestion! We have updated the empirical results in our revision, where all evaluations are averaged from 5 random seeds. The statistics variance are shown below. The rankings and observations are still consistent with our previous conclusions.
>
> | Method | EVR | | | GIER | | | MA5k | | | MagicBrush | | | |
> | :-- | :-- | :-- | :-- | :-- | :-- | :-- | :-- | :-- | :-- | :-- | :-- | :-- | :-- |
> | **Zero-shot** | **L1↓** | **DINO↑** | **CVS↑** | **L1↓** | **SSIM↑** | **CVS↑** | **L1↓** | **SSIM↑** | **LPIPS↓** | **L1↓** | **DINO↑** | **CVS↑** | **CTS↑** |
> | InsPix2Pix | 0.189±0.013 | 67.82±3.09 | 81.38±3.58 | 0.144±0.007 | 57.51±1.51 | 86.63±1.44 | 0.176±0.005 | 58.92±1.15 | 0.359±0.071 | 0.101±0.009 | 71.46±3.12 | 85.22±3.59 | 29.34±0.30 |
> | LGIE | **0.159**±0.017 | 69.71±2.89 | **82.04**±1.66 | 0.152±0.014 | 56.86±2.37 | 86.99±2.14 | 0.144±0.017 | 64.60±1.88 | 0.327±0.006 | 0.084±0.041 | 80.90±2.07 | 88.87±1.79 | 30.10±0.29 |
> | MGIE | 0.163±0.008 | **71.49**±3.83 | 81.73±2.99 | **0.135**±0.011 | **59.24**±0.72 | **88.59**±0.75 | **0.133**±0.018 | **66.25**±2.41 | **0.298**±0.053 | **0.082**±0.016 | **82.22**±2.70 | **91.14**±4.06 | **30.40**±0.31 |
>
> > **Q) evaluation metrics**
>
> Yes. All ground-truth goal images are available in EVR, GIER, MA5k, and MagicBrush. For Photoshop-style modification (EVR and GIER), we adopt semantic-level visual similarity DINO and CVS. Since there are many photo optimization cases in GIER, we also consider SSIM. MA5k aims to adjust the contrast/brightness/saturation, where DINO and CVS cannot clearly tell the difference; Hence we leverage the widely-used style distance LPIPS. As we have the ground-truth description in MagicBrush, we follow the additional CTS to evaluate the alignment between goal captions and edited images.
>
> > **Q) FID score**
>
> We calculate the FID score between ground-truth and edited images in Sec. A (Evaluating Image Editing via FID). However, the differences are all pretty limited. Since most editing results still resemble the original input images, it is difficult for FID to determine their authenticity. These results indicate that FID is insufficient to compare the quality of image editing.
>
> | Method | Zero-shot | | | | Fine-tuned | | | |
> | :-- | :-- | :-- | :-- | :-- | :-- | :-- | :-- | :-- |
> | | **EVR** | **GIER** | **MA5k** | **MagicBrush** | **EVR** | **GIER** | **MA5k** | **MagicBrush** |
> | InsPix2Pix | **6.19** | **5.61** | 5.91 | 5.69 | **5.31** | **5.31** | **5.30** | 5.64 |
> | LGIE | 6.67 | 5.69 | 5.80 | **5.31** | 5.32 | 5.42 | 5.59 | 5.48 |
> | MGIE | 6.45 | 5.64 | **5.48** | 5.61 | 5.53 | 5.59 | 5.41 | **5.42** |
>
> > **Q) training cost**
>
> We updated our training cost in Sec. B (Training Cost). Our MGIE training requires 26 epochs to converge, and InsPix2Pix has 20 epochs (from their released checkpoint). Both MGIE and InsPix2Pix take a similar 1.6 hours per epoch on our node (8 NVIDIA A100 GPUs), where the overall training can be done in two days.

---

> > ### Author Response · Authors · 2023-11-21
> >
> > Dear Reviewer BZAL, we are thankful for your review. As the rebuttal deadline is coming to an end, please let us know if your concerns are well addressed. We are happy to provide further clarification.

---

> > > ### Comment · Reviewer_BZAL · 2023-11-21
> > >
> > > I thank the authors for their rebuttal, thorough efforts to address all reviewers concerns, and improvements to the paper's clarity. Most of my concerns were addressed, and I recommend acceptance of the manuscript.

---

### Meta-Review · Area_Chair_AfWC · 2023-12-10

**Metareview:**

The submission introduces multimodal large language model (MLLM)-guided image editing. The key idea is that image editing instructions provided by humans are often short and assume implicit knowledge of the scene. MLLMs can expand the instruction using their comprehensive knowledge of language and vision acquired during training. These expressive instructions can help provide better editing capabilities.
The problem tackled is interesting and of practical value. The qualitative and quantitative results provided were appreciated by the reviewers.
Some minor weaknesses were raised by the reviewers, which were sufficiently addressed in the rebuttal phase.

**Justification For Why Not Higher Score:**

The submission is not exceptionally novel worthy enough to be an oral presentation. Similar works have been previously published. This one provides incremental but useful updates.

**Justification For Why Not Lower Score:**

The submission received unanimously positive reviews, with an average of 7.

---

### Decision · Program_Chairs · 2024-01-16

Accept (spotlight)